# Quantitative comparison of measured and simulated $O_4$ absorptions for one day with extremely low aerosol load over the tropical Atlantic

Thomas Wagner[1], Steffen Dörner[1], Steffen Beirle[1], Sebastian Donner[1], Stefan Kinne[2]

[1]Satellite Remote Sensing Group, Max Planck Institute for Chemistry, Mainz, Germany
[2]Max-Planck Institute for Meteorology, Hamburg, Germany

*Correspondence to*: Thomas Wagner (thomas.wagner@mpic.de)

**Abstract**. In this study we compare measured and simulated $O_4$ absorptions for conditions of extremely low aerosol optical depth (between 0.034 to 0.056 at 360nm) on one day during a ship cruise in the tropical Atlantic. For such conditions, the uncertainties related to imperfect knowledge of aerosol properties don't significantly affect the comparison results. We find that the simulations underestimate the measurements by 15% to 20%. Even for simulations without any aerosols the measured $O_4$ absorptions are still systematically higher than the simulation results. The observed discrepancies can not be explained by uncertainties of the measurements and simulations and thus indicate a fundamental inconsistency between simulations and measurements.

## 1 Introduction

Remote sensing measurements of the atmospheric absorption of the oxygen dimer $(O_2)_2$ are often used to derive properties of aerosols and clouds. The atmospheric concentration of $(O_2)_2$ (in the following referred to as $O_4$) varies only slightly with temperature, pressure and humidity (aside from the dependence on altitude). Thus deviations from the $O_4$ absorptions for clear sky conditions indicate changes of the atmospheric radiative transfer, e.g., due to clouds and aerosols. In recent years, inconsistencies between the measured atmospheric $O_4$ absorption and radiative transfer simulations were detected for Multi-AXis-DOAS (MAX-DOAS) observations. MAX-DOAS instruments measure scattered sun light under different, mostly slant elevation angles (Hönninger and Platt, 2002). Several studies found that a scaling factor (SF<1) had to be applied to the observed atmospheric $O_4$ absorptions in order to bring them into agreement with radiative transfer simulations (e.g. Wagner et al., 2009; Clémer et al. 2010). Other studies, however, did not find the need to apply such a scaling factor (e.g. Spinei et al., 2015; Ortega et al., 2016). A more detailed discussion and overview on existing studies of both groups is provided in Wagner et al., 2019. One major difficulty in the quantitative interpretation of these comparisons is that usually the atmospheric aerosol properties are not well known (e.g. the vertical extinction profile and/or the optical properties). And even if they were known, it is still a challenge to accurately represent them in atmospheric radiative transfer simulations.

In this study, we minimise these difficulties by using atmospheric observations in the presence of very low aerosol loads. During a ship campaign across the tropical Atlantic, very low aerosol optical depth (AOD) was observed on one day (2 May 2019). At 360 nm (the wavelength at which we analyse the atmospheric $O_4$ absorption), the AOD ranged from 0.034 to 0.056, which is an order of magnitude lower than the optical depth of molecular Rayleigh scattering.

Like in previous studies, we compare the observed atmospheric $O_4$ absorption with the results of radiative transfer simulations. Information about the aerosol properties is derived from sun photometer measurements in combination with ceilometer measurements. Also in our study, considerable uncertainties about the aerosol vertical profile and the aerosol optical properties exist. However, these uncertainties are less important for the interpretation of the comparison results than in previous studies because of the low AOD, and we find large discrepancies between the measured and simulated $O_4$ absorptions.

The paper is organised as follows: In section 2, an overview on the ship campaign and the instruments used in this study is given. Sections 3 to 5 describe the spectral analysis, the cloud classification, and the calculation of the $O_4$ profile. In section

6, the radiative transfer simulations and the extraction of the aerosol extinction profiles are presented. Section 7 presents the comparison results, and section 8 the summary and conclusions.

## 2 Overview on the ship campaign and the instruments used in this study

The MAX-DOAS measurements were carried out during a cruise (MSM82/2) of the German research vessel RV Maria S. Merian (https://www.ldf.uni-hamburg.de/merian.html) from Montevideo (Uruguay) to Las Palmas (Spain) from 26 April 2019 to 14 May 2019 (see Fig. 1). More details on the ship cruise MSM82/2 can be found in Krastel et al. (2019). In this study, we focus on one day with particularly low AOD (2nd May), which is marked in Fig. 1.

The MAX-DOAS instrument was mounted above the ship's bridge at about 20m altitude above sea level. The telescope was aligned in the driving direction of the ship (Fig. 2).

### 2.1 MAX-DOAS instrument

The MAX-DOAS instrument is a so-called Tube MAX-DOAS instrument which was developed and built by the electronic workshop of the Max Planck Institute for Chemistry in Mainz (Donner, 2016). It consists of two major parts: the telescope unit and the spectrometer unit. The telescope unit is mounted outside on the railing of the ship. The spectrometer unit is located inside the ship. Besides the spectrometer it also contains a peltier cooling element which stabilises the spectrometer temperature at 15 °C. Both units are connected via a quartz glass fibre bundle and electric cables. The telescope unit is equipped with a gyroscope to stabilise the elevation angles by continuously adjusting the motor position with an accuracy of ±0.1°.

The spectrometer is an Avantes ULS2048x64-USB2. It covers the spectral range from 299.4 nm to 463.1 nm with a spectral resolution between 0.52 and 0.54 nm as described by the full width half maximum (FWHM). Spectra are measured with an integration time of 1 min at the following elevation angles: -2°, -1°, -0.5°, 0°, 0.5°, 1°, 2°, 3°, 4°, 5°, 6°, 8°, 10°, 15°, 30°, 90°. Note that in this study only measurements with positive elevation angles are used. One elevation sequence is completed within about 21 min. Dark current and offset spectra are taken during night and are used to correct the measured spectra before the spectral analysis.

### 2.2 Sun photometer

A MICROTOPS II sunphotometer provided atmospheric totals on aerosol and water vapor. The instrument, when directed towards the sun (in a handheld operation), captures via diodes the solar intensity in five sub-spectral bands near wavelengths of 380, 440, 675, 870 and 940 nm. In combination with the larger reference solar intensity at the top of the atmosphere - using time and (GPS-provided) position data - sun-photometer measurements define the atmospheric attenuation at these solar sub-spectral bands. Four spectral bands (near 380, 440, 675 and 870 nm) sample in trace-gas poor regions, while one spectral band (near 940nm) is strongly affected by water vapor absorption. In the absence of clouds, the solar attenuations in the four trace-gas poor bands can be linked to aerosol - after (surface air pressure defined) contributions from air-molecule (Rayleigh) scattering have been removed. Hereby, the aerosol associated attenuations are quantified by the (vertically normalized) aerosol optical depth (AOD). As the instrument offers AOD values simultaneously at four different solar wavelengths, the typical aerosol particle size is revealed and even AOD contributions from sub-micrometer (mainly from pollution and wildfire) and super-micrometer size aerosol particles (mainly from dust and seasalt) can be distinguished. The determination of the atmospheric water vapor is based on the differential absorption between 870 and 940 nm attenuation data. Any quality measurement usually relies on many repeated samples in order to identify and remove poor data associated

with sun-view contamination by clouds and/or inaccurate orientations of the instrument into the sun (which is done manually with the help of a pointing device). NASA's Aeronet sub-group of the Maritime Aerosol Network (MAN, Smirnov et al., 2009) provided the calibrated instrument for the cruise and also stores cruise data at https://aeronet.gsfc.nasa.gov/new_web/cruises_new/Maria_Merian_19_0.html.

The AOD at 360 nm for 2 May 2019 is shown in Fig. 3. Other results from the sun photometer measurements (AODs at different wavelengths, Angström exponents, and fine and coarse mode AOD) are presented in Fig. A3).

It should be noted that the uncertainties of the last AOD measurement on 2 May 2019, 19:26, are rather large because of the high SZA of 85°. In particular it was found that for that measurement the AOD from the fully processed sun photometer data (Fig. A3) was about 30% larger than the AOD of the initial retrieval (Fig. 3), while the results for all other measurements are almost identical. The radiative transfer simulations presented below for the last elevation sequence (19:06 to 19:25) are based on the initial (low) AOD values, which are in agreement with AOD measurements 20 minutes earlier. Nevertheless, the comparison results for this last elevation sequence should be treated with caution because of the large uncertainties of the corresponding AOD measurement.

**2.3 Ceilometer**

The Jenoptik 15K ceilometer of the MPI-M is a simple laser system operating at 1064nm, at an invisible trace-gas free near-IR wavelength. Laser impulses are sent upward into the atmosphere and based on strength and delay of backscattered return signals altitude positions for atmospheric aerosol and clouds are derived. Due to their stronger backscatter at optically thicker media, such as clouds, overhead cloud base altitudes are well captured. However, as laser light strongly attenuates in optically thicker media, no information above a cloud base is possible. Vertical profiles of aerosol for clouds-free views (and below clouds) are possible up to about 7km in altitude during the night but only up to about 4km in altitude during the day, due to scattering noise by sunlight. No useful aerosol profiling is possible near the surface (e.g. lower 300m), because signal sender and receiver are not at the identical location. Recorded ceilometer data of the cruise are accessible via an anonymous ftp-site at ftp://ftp-projects.zmaw.de/aerocom/ships/ceilometer_MSM/.

**3 Spectral analysis**

The spectral analysis is performed following mostly the settings suggested by Wagner et al. (2019). The spectral range from 352 to 385 nm is chosen, which contains two $O_4$ absorption bands. Note that in Wagner et al. (2019) the wavelength range 352 – 387 nm was used. Here we restricted it to 352 - 385 nm, because for some measurements (not on 2 May 2019) large spectral structures were found > 385 nm). For 2 May 2019, almost identical results (differences < 1%) were found for both spectral ranges. The details of the analysis are given in Table 1. Fig. A1 (left) presents an example of the spectral analysis as used in this study. In addition to the other cross sections, also a $H_2O$ cross section (Polyansky et al. 2018) is included. The reason for including a $H_2O$ cross section as well as the effect of including a second $O_4$ cross section are discussed in appendix A1.

The results of the spectral analysis represent the integrated trace gas concentration along the atmospheric light path, the so-called slant column density (SCD). For $O_4$ the SCD is expressed with respect to the square of the $O_2$ concentration (see Greenblatt et al., 1990). Thus, the unit of the $O_4$ SCD is molec²/cm⁵. For the analysis of the measured spectra, a so-called Fraunhofer reference spectrum is used. In this study, the Fraunhofer reference spectrum is calculated as the average of the zenith spectra before and after the chosen elevation sequence, weighted by the time of the selected measurement from that elevation sequence. Before performing the spectral analysis, these sequential Fraunhofer reference spectra are fitted to a ‚universal' Fraunhofer reference spectrum (29 April 13:43, SZA: 44.8°, elevation angle: 90°) to transfer the spectral

calibration of the universal Fraunhofer reference spectrum to the sequential Fraunhofer reference spectra. The universal Fraunhofer reference spectrum was calibrated using a high resolved solar spectrum.

Since the Fraunhofer reference spectrum also contains atmospheric trace gas absorptions, the output of the spectral analysis represents the difference between the SCDs of the selected non-zenith spectrum and the Fraunhofer reference spectrum, the so-called differential SCD (or dSCD).

The typical fit error of the derived $O_4$ dSCD is between $2 \cdot 10^{41}$ molec$^2$/cm$^5$ and $4 \cdot 10^{41}$ molec$^2$/cm$^5$. Depending on the magnitude of the retrieved $O_4$ dSCD this corresponds to relative errors between 1 and 4 %.

**4 Cloud detection using the MAX-DOAS measurements**

Although during most of the afternoon on 2 May clear sky conditions prevailed, also some scattered clouds were present. They were e.g. detected by the ceilometer in zenith direction (see Fig. 3). In order to derive information about possible cloud contamination for the individual MAX-DOAS measurements, the MAX-DOAS measurements themselves were used for the detection of cloud contamination, similar as in Wagner et al., (2014, 2016). Figure A4 in the appendix shows the time series of the retrieved $O_4$ dSCDs on 2 May for the different elevation angles. During the morning the $O_4$ dSCDs show strong variability caused by the presence and variability of clouds as also seen in the ceilometer data (Fig. 3). During the afternoon, for most of the time, smooth variations of the $O_4$ dSCDs are found indicating clear sky conditions. However, for some times and elevation angles, also small systematic deviations (usually reductions) of the $O_4$ dSCDs occur, which are caused by scattered clouds. During periods without any cloud contamination, the temporal variability of the retrieved $O_4$ dSCDs is rather small (scatter of the $O_4$ dSCDs is typically $\leq 5 \cdot 10^{41}$ molec$^2$/cm$^5$). Measurements with deviations $> 10^{42}$ molec$^2$/cm$^5$ compared to the extrapolated $O_4$ SCDs from the smooth (cloud-free) neighboring measurements are thus flagged as cloud-contaminated. From the selected 11 elevation sequences during the mainly cloud-free periods in the afternoon of 2 May, 7 are found to be completely free of cloud contamination.

**5 Calculation of the $O_4$ profile and $O_4$ VCD**

The $O_4$ height profile and VCD for 2 May 2019 are calculated from vertical profiles of temperature and pressure. Also the effect of the atmospheric humidity is accounted for. For the profiles of temperature, pressure and atmospheric humidity we used the results from the ECMWF ERA-Interim data set (Berrisford et al., 2011) for 2 May 2019. From the temperature and pressure profiles the air concentration [air] is calculated. Then the $O_2$ concentration $[O_2]$ is derived according to the following equation:

$$[O_2] = [air] \cdot M_{O2} \cdot (1 - M_{H2O}) \qquad (1)$$

Here $M_{H2O}$ is the mixing ratio of water vapour taken from the ERA interim data. For the dry air mixing ratio of $O_2$ ($M_{O2}$) a value of 21% is assumed. The $O_4$ concentration is then represented by the square of the $O_2$ concentration (Greenblatt et al., 1990). To derive the $O_4$ VCD, the $O_4$ concentration is vertically integrated between the surface and 30 km with a vertical resolution of 20 m.

The temperature and pressure from the ECMWF ERA-Interim data set at the surface are also compared to the in situ measurements on the ship. It is found that the ECMWF temperature is slightly lower (-0.7 K) and the ECMWF pressure is slightly higher (+2 hPa) than the corresponding in situ measurements, see Fig. A5 in the appendix. Therefore, we repeated our calculations of the $O_4$ profiles by shifting the ECMWF values for the whole profiles by +0.7 K and – 2 hPa. The resulting change of the $O_4$ VCD is rather small (+0.3 %). The derived $O_4$ VCD for the modified profile is $(1.245 \pm 0.25)$

$\cdot 10^{43}$molec$^2$/cm$^5$. The most probable reasons for the discrepancies are originating from the rather coarse horizontal (~80 km) and temporal (6 h) resolution of the ECMWF interim data set: First, the given model data is the average for the modelled box. Moreover, the simulation uncertainties are increased for parameterized subscale processes (e.g. wave motion) which do affect the in-situ measurements.

To estimate the uncertainty of the derived $O_4$ VCD the temperature and pressure of the whole profiles are varied by ±2 K and ±2 hPa, respectively. The resulting changes of the $O_4$ VCDs are ±1.5 % and ±0.9 %, respectively. In addition, assuming an uncertainty of the atmospheric humidity profile of 30% leads to an uncertainty of the derived $O_4$ VCD of 0.9 %. Thus, we estimate the total uncertainty of the $O_4$ VCD to ±2 %.

Finally, a subtle detail should be mentioned: the integration of the $O_4$ VCD was performed starting from sea level, while the instrument was located about 20 m above sea level. This rather small difference would result in a reduction of the $O_4$ VCD by 0.4 %. However, this effect is considered in exactly the same way in the radiative transfer simulations, where the instrument was also put at an altitude of 20 m, while the $O_4$ AMFs are calculated for the $O_4$ column starting from sea level. Thus it is a consistent procedure to use the $O_4$ VCD integrated from sea level for the conversion of the measured $O_4$ dSCDs into $O_4$ dAMFs.

**6 Radiative transfer simulations**

$O_4$ dSCDs are calculated using the full spherical Monte Carlo radiative transfer model MCARTIM (Deutschmann et al. 2011). For the simulations, the profiles of temperature, pressure, and $O_4$ as described in section 5 are used. The vertical resolution was set to 20 m close to the surface and increases with altitude (see Table 2). The surface albedo was set to 0.05. The value of 5% was chosen to be consistent with the MAPA inversions, and because it is appropriate for many parts of the global ocean. However, by having a closer look at maps of albedo (Kleipool et al., 2008) and chlorophyll content (e.g. from the NASA Earth Observatory: https://earthobservatory.nasa.gov/global-maps/MY1DMM_CHLORA), we found that at the specific location of the measurements, very clear waters exist, for which the surface albedo is typically higher (about 7 to 8 %). The presence of very clear waters was also supported by the in situ chlorophyll measurements aboard the ship. We therefore made additional radiative transfer simulations using a surface albedo of 8 %. We found that the obtained $O_4$ dAMFs were almost identical with those for 5 % surface albedo (differences <1%). The reason for the good agreement is that the effect of the surface albedo is similar for the $O_4$ AMFs for different elevation angles. Thus the effect of varying surface albedo almost cancels out.

The simulations were performed for the exact SZA and relative azimuth angles of the individual measurements. From the obtained $O_4$ AMFs, the corresponding $O_4$ dAMFs are calculated by subtracting the simulated $O_4$ AMFs for the zenith viewing direction. To achieve best consistency with the measurements, for the simulation of the zenith measurements (interpolated between the zenith observations before and after the sequence) the SZA and relative azimuth angle for the exact time of the non-zenith measurements are also used for the simulations of the zenith measurements. The temporal evolution of the SZA and relative azimuth angle for 2 May are shown in Fig. A6 in the appendix.

It should be noted that it is important to use a consistent treatment of the SZA and relative azimuth angles in the simulations and measurement analyses. Especially the choice of the Fraunhofer reference spectra is important. If e.g. either zenith measurements before or after the selected elevation sequence are used as reference spectra, systematic deviations of the retrieved $O_4$ dSCDs of up to 10% can occur (see Fig. A7 in the appendix).

$O_4$ dAMFs are simulated for two aerosol extinction profiles as well as for a pure Rayleigh atmosphere. For the extraction of the aerosol extinction profiles, the observations by the sun photometer and the ceilometer were used (see section 6.1). For the simulations including aerosols, the phase function is represented by a Henyey-Greenstein (HG) parameterisation with an asymmetry parameter of 0.68. The single scattering albedo was set to 0.95. Variations of these properties lead to changes of

the simulated $O_4$ dSCDs by up to $\pm 3$ % (see Fig. A8 in the appendix). These rather low uncertainties are related to the low AOD on 2 May 2019. For measurements with higher aerosol loads, the corresponding uncertainties are usually much larger (e.g. Wagner et al., 2019). Here it should be noted that the HG phase function model is a rather simplified approximation for true aerosol phase functions. Thus especially for measurements with small scattering angles (e.g. around noon on 2 May 2019) the uncertainties of the RTM simulations might be larger.

**6.1 Extraction of the aerosol extinction profiles**

Figure 4 presents the hourly averaged and range corrected ceilometer backscatter profiles for three periods in the afternoon on 2 May 2019 without cloud contamination. In a first order approximation, these backscatter profiles are proportional to the aerosol extinction. Thus together with the total AOD from the sun photometer measurements, the aerosol extinction profiles can be determined. However, ceilometer measurements are affected by several instrumental limitations, which complicate the direct conversion to aerosol extinction profiles:

a) Due to the missing overlap between the outgoing beam and the field of view of the detector, the sensitivity of the ceilometer is very low for altitudes below 500 m. Thus for this altitude range, no information on the aerosol extinction can be derived from the ceilometer measurements.

b) In spite of the long averaging period, still strong noise appears for altitudes above 3 km.

Due to these limitations, the ceilometer profiles can only be used for a restricted altitude range. In the following we used the ceilometer profiles for the altitude range between 500 m and about 7 to 9 km. Between 500 m and 3000 m, averages for 100 m layers are calculated. Below 500 m, the values at 500 m are either set constant for the layer below, or are linearly extrapolated from the ceilometer data between 500 m and 800 m (similar as in Wagner et al., 2019). Since between 3 km and 10 km the noise increases strongly, a third order polynomial was fitted to the ceilometer data in that height range. The polynomial values are used for the altitude range for which positive values are obtained. Between 7 and 9 km the polynomial values for the three profiles cross zero. Above these altitudes, the profile values are set to zero. These extraction steps are illustrated in Fig. A9 in the appendix. Here it should be noted that the exact choice of the altitude, at which the extinction is set to zero, has negligible influence of the simulated $O_4$ dAMFs.

Before the backscatter profiles are normalised with the total AODs measured by the sun photometer, the stratospheric part of the total aerosol profile has to be added. This step is usually not important, because in more polluted areas the total AOD is clearly dominated by the tropospheric part. However, for our study, the total AOD is so low that the stratospheric part constitutes a substantial fraction (up to 25 %) of the total AOD. Thomason et al. (2018) report the stratospheric AODs in the Tropics at 525 nm to be about 0.005 to 0.006. Assuming an Angström exponent of 2 (e.g. Malinina et al., 2019) the corresponding AOD at 360 nm is estimated to around 0.011 and 0.013. In the following we used a value of 0.012. Here it should be noted that the Angström exponent for stratospheric aerosols is usually derived for wavelengths at and above 525 nm. Thus it is not clear how representative the used value of 2 is also for shorter wavelengths. To estimate the uncertainties of the simulated $O_4$ dSCDs related to the uncertainty of the Angström exponent, we performed additional radiative transfer simulations assuming a stratospheric AOD of 0.008 (corresponding to an Angström exponent of about 1). We found that the $O_4$ dSCDs differ from those for a stratspheric AOD of 0.012 by less than 1%.This stratospheric AOD (0.012) is then subtracted from the total AOD (Fig. 3) measured by the sun photometer. Then the tropospheric aerosol profiles (as described above, see also Fig. A9) are normalised by the resulting tropospheric AOD. Finally, the stratospheric extinction profile is added to the normalised tropospheric aerosol extinction profiles. For the stratospheric extinction profile we used a simplified shape with an AOD of 0.012. Here it is important to note that the details of the extinction profile in the upper troposphere and stratosphere are not critical. For example, the simulated $O_4$ dAMFs using aerosol profiles with or without the

stratospheric part are almost the same. The final aerosol extinction profiles used for the RTM simulations are shown in Fig. 4.

It should be noted that the aerosol properties can change with altitude. Thus the relative profile shape measured at 1064 nm might differ from the aerosol extinction profile at 360 nm. In order to estimate the effect of the varying aerosol profiles at both wavelengths, we performed additional radiative transfer simulations using modified tropospheric aerosol profiles. The aerosol extinction in the lowest 1000 m of the extracted profiles was changed by +/-20% and the free tropospheric part above was adjusted to keep the total AOD unchanged. The resulting $O_4$ dAMFs were almost unchanged for elevation agles >4°. For

lower elevation angles, the changes were found to be +/-2%.

**6.2 Calculation of effective temperatures for the $O_4$ absorption**

Since the temperature of the troposphere decreases with altitude, and the $O_4$ absorption cross section depends on

temperature, the retrieved $O_4$ dSCDs might deviate from the true $O_4$ dSCDs (the integrated $O_4$ concentration along the atmospheric light paths), because only one $O_4$ cross section for a fixed temperature is used in the spectral analysis. Thus, before the $O_4$ dAMFs from the measured spectra are compared to those from the radiative transfer simulations, the effect of the temperature dependence of the $O_4$ absorption has to be investigated.

The effective temperature of the $O_4$ measurements is calculated according to:


$$T_{eff,\alpha} = \frac{\sum_z [O_4]_z \cdot (bAMF_{z,\alpha} - bAMF_{z,90°}) \cdot T_z}{\sum_z [O_4]_z \cdot (bAMF_{z,\alpha} - bAMF_{z,90°})} \qquad (2)$$

Here $[O_4]_z$ represents the $O_4$ concentration at altitude z, $bAMF_{z,\alpha}$ the box-AMF for elevation angle $\alpha$ at altitude z, and $T_z$ the temperature at altitude z. $T_{eff,\alpha}$ is the effective temperature for the measured $O_4$ dSCD at elevation angle $\alpha$.

Equation 2 is applied for each individual measurement, the results are shown in Fig. A10. The effective temperatures range from 276 K to 299 K. They depend systematically on the elevation angle and SZA. Measurements at low elevation angles are most sensitive for the layers near the surface, at which the highest temperatures occur. Measurements at high SZA (towards the end of the considered time period) have higher sensitivities for higher atmospheric layers with colder temperatures. Both dependencies are well represented by the results shown in Fig. A10.

To correct the effect of the temperature dependence, the correction factors presented in Fig. 13 in Wagner et al. (2019) are applied to the $O_4$ dSCDs retrieved with the $O_4$ cross section for 293 K. The corrected $O_4$ dSCDs differ by up to a few percent (between –2 % and +7 %) from the original $O_4$ dSCDs. In Fig. A11 in the appendix the effect of the temperature correction is shown for two selected elevation sequences. For the comparison with the radiative transfer simulations the temperature-corrected $O_4$ dSCDs (or dAMFs) are used.


**7 Comparison results**

**7.1 Direct comparison between measurements and RTM results**

In Fig. 6 the $O_4$ dAMFs derived from the MAX-DOAS measurements are compared to those obtained from the radiative transfer simulations for elevation sequences not affected by clouds (similar comparisons for the sequences with cloud-contaminated measurements are shown in Fig. A12 in the appendix).

In the left part of Fig. 6, the results from radiative transfer simulations without aerosols are shown. Here, for almost all cases, the measured $O_4$ dAMFs are systematically larger than the simulated $O_4$ dAMFs. This is an important finding, because

especially for the low elevation angles, the presence of aerosol scattering leads to a decrease of the $O_4$ dSCDs. Thus the simulations for a pure Rayleigh atmosphere represent an upper limit of the achievable $O_4$ dSCDs. Only for cloudy cases with a high probability for multiple scattering events higher $O_4$ dSCDs could occur, but such conditions can be ruled out here because of the absence of thick and vertically extended clouds. Thus the overestimation of the simulated $O_4$ dSCDs for a pure Rayleigh atmosphere by the measured $O_4$ dSCDs indicates a fundamental inconsistency between measurements and

simulations. Similar results are found for the elevation sequences with cloud contamination (Fig. A12 in the appendix).

In the right part of Fig. 6, simulation results for the aerosol profiles extracted in section 6.1 are shown. Note the separate y-axes for the simulated $O_4$ dAMFs on the right side, for which the maxima are chosen to achieve best agreement between the measured and simulated $O_4$ dAMFs. The exact values of the axis maxima were determined by fitting the measured $O_4$ dAMFs to the simulated $O_4$ dAMFs for elevation angles >4°. For these elevation angles the simulation results for the

different profile shapes below 500m) are almost the same. Good qualitative agreement between measurements and simulation is found, especially for the aerosol profiles with constant extinction below 500 m. However, the absolute values differ strongly. The ratios between measured and simulated $O_4$ dAMFs are found to between 0.8 and 0.86. Again, similar results are found for the elevation sequences with cloud contamination (Fig. A12 in the appendix).

The scaling factors derived from this comparison between measured and simulated $O_4$ dSCDs are presented as blue data

points in Fig. 7.

It should be noted that during the entire ship cruise, only during the beginning of 3 May 2019, similarly low (but still larger) AOD were measured as on 2 May 2019. We compared the measured $O_4$ dAMFs for the first two elevation sequences on 3 May with radiative transfer simulations. For that comparison we only made simulations for an aerosol-free atmosphere in order to limit the effort (and also because of the rapid temporal variation of the AOD during that time period). The results

(see Fig. A13) are similar to those on 2 May 2019: Except for the cloud contaminated measurements, the simulations are smaller than the measurements.

### 7.2 Profile inversion with MAPA

We also applied our profile inversion algorithm, the Mainz profile algorithm (MAPA, Beirle et al., 2019), to the measured $O_4$ dAMFs. For that purpose, a new MAPA LUT had to be created, because the lowest AOD in the original LUT (0.05) is larger than all AODs observed on 2 May 2020. The new LUT includes AOD values from zero to 0.1 in steps of 0.02. MAPA provides the option to apply a fixed user-defined scaling factor, or to determine a scaling factor yielding best match between forward model and measurement during profile inversion.

In Fig. A14 in the appendix the retrieved extinction profiles are shown for different scaling factors. Here it should be noted that the individual measurements (not the sequences) with cloud contamination were skipped before the profile inversion. Only profiles with either ‚valid' or ‚warning' flags are shown (profiles with ‚error' flags are not shown). In Fig. A15 in the appendix the retrieved AODs for the different scaling factors are compared to the tropospheric AODs from the sun photometer measurements (stratospheric AOD of 0.012 was subtracted). Also the RMS between the measured and simulated

$O_4$ dAMFs are shown (right). The colour of the MAX-DOAS inversion results indicates the quality of the profile inversion.

Most valid profiles are obtained for scaling factors between 0.80 and 0.90, or for a free fitted (variable) scaling factor. For the inversions with larger scaling factors, rather high RMS are found. For most cases, the retrieved AODs are smaller than those measured by the sun photometer. However, here it should be noted that for these low aerosol extinctions, the information content of the measurements is probably too low to constrain the aerosol extinction profiles, especially for high

altitudes. Thus also the retrieved AOD values are very unstable (see Fig. A15)..

The obtained scaling factors are shown in Fig. 7. Overall good agreement between both comparison methods is found. For all elevation sequences, values of the scaling factor < 1 are found. For the direct comparison, the difference from unity is mostly larger than 15 % and can thus not be explained by the uncertainties of the measurements and simulations, which are summarised in table 3..

## 8 Conclusions

We compared measured and simulated $O_4$ absorptions for one day with very low aerosol optical depth. For such conditions, the uncertainties caused by imperfect knowledge of the aerosol properties play a smaller role than for comparison under more polluted conditions.

One important result of the comparison was that for all measurements, the observed $O_4$ absorption was higher than the simulation results for an atmosphere without aerosols. In the absence of optically thick clouds, the simulated $O_4$ dAMFs for an atmosphere without aerosols constitutes an upper limit, since especially for the low elevation angles the inclusion of aerosols leads to a decrease of the $O_4$ absorption. The observed discrepancies thus indicate a fundamental inconsistency between simulations and measurements.

The measured $O_4$ absorptions are also compared to simulations including aerosol extinction profiles. The aerosol extinction profiles were constrained by measurements of the sun photometer, the ceilometer and a climatology of stratospheric aerosols. Again, a large discrepancy was found for the absolute values. However, for the relative dependence of the $O_4$ dAMFs on the elevation angle good agreement could be achieved. For each elevation sequence, the ratio of simulated and measured $O_4$ dAMFs was calculated. For that purpose the elevation angles >4° were used, for which the $O_4$ dAMFs are almost insensitive to the profile shape in the lower atmospheric layers. For all elevation sequences, ratios of 0.85 or less were found. Similar ratios were also obtained from the application of our profile inversion algorithm (MAPA) to the measurements. The observed discrepancies cannot be explained by the uncertainties of measurements and/or simulations. Here it is important to note that in the spectral analysis, we explicitly corrected for the (small) temperature dependence of the atmospheric $O_4$ absorption.

Our results indicate that something fundamental is missing/wrong in either the radiative transfer simulations or the spectral analysis of the atmospheric $O_4$ absorptions. We did not find a clear reason for the discrepancies. One possible reason for the discrepancies could be a systematically too small $O_4$ absorption cross section.

We recommend that similar studies under extremely low aerosol load should be made at different locations and seasons. Also $O_4$ absorptions at different wavelengths should be investigated.

**Author contributions**

T. Wagner performed the measurements, data analysis and prepared the manuscript. S. Dörner prepared the MAX-DOAS instrument and extracted the ERA interim data. S. Dörner and S. Donner contributed to the MAX-DOAS operation and data analysis. S. Beirle performed the MAPA profile inversions. S. Kinne operated the sun photometer and ceilometer.

**Acknowledgements**

The scientific party of RV MARIA S. MERIAN Cruise MSM82/2 gratefully acknowledges the very friendly and most effective cooperation with Captain Maaß and his crew. Their great flexibility and their perfect technical assistance substantially contributed to make this cruise a scientific success. We also appreciate the valuable support by the Leitstelle Deutsche Forschungsschiffe (German Research Fleet Coordination Centre) at the University of Hamburg. The expedition was funded by the Deutsche Forschungsgemeinschaft.

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

**Tables**

**Table 1: Settings for the DOAS analysis of $O_4$**

| Parameter | Value, Remark / Reference |
|---|---|
| Spectral range | 352 – 387 nm |
| Degree of DOAS polynomial | 5 |
| Degree of intensity offset polynomial | 2 |
| Fraunhofer reference spectrum | Interpolated between 90° measurement before and after each elevation sequence |
| Wavelength calibration | Fit to high resolution solar spectrum using Gaussian slit function |
| Shift / squeeze | The measured spectrum is shifted and squeezed against all other spectra |
| Ring spectrum 1 | Normal Ring spectrum calculated from measured zenith spectrum |
| Ring spectrum 2 | Ring spectrum 1 multiplied by $\lambda^{-4}$ (Wagner et al. (2009) |
| $O_3$ cross section | 223 K, Bogumil et al. (2003) |
| $NO_2$ cross section | 294 K, Vandaele et al. (1997) |
| $H_2O$ cross section | 293 K, Polyansky et al. (2018) |
| $O_4$ cross section | 293 K, Thalman and Volkamer (2013) |


**Table 2: Vertical resolution used for the radiative transfer simulations**

| Altitude range [km] | Vertical resolution [km] |
|---|---|
| 0 - 0.5 | 0.02 |
| 0.5 - 2 | 0.1 |
| 2 – 12 | 0.2 |
| 12 – 25 | 1 |
| 25 – 45 | 2 |
| 45 - 100 | 5 |






**Table 3: Uncertainties related to the different analysis steps**

| Spectral analysis | | |
|---|---|---|
| **Effect** | **Magnitude** | **Reference** |
| Spectral fit | 1 - 4% | Result of spectral fit |
| Temperature dependence | 1.5% | Wagner et al., 2019 |
| Fit paramaters | 3.5% | Appendix A1, and Wagner et al., 2019 |
| Total | 4 – 5.5% | |
| | | |
| **RTM without aerosols** | | |
| $O_4$ profile | 1% | Wagner et al., 2019 |
| albedo | 1% | Section 6 |
| RTM general | 1% | Wagner et al., 2019 |
| total | 2% | |
| | | |
| **RTM with aerosols** | | |
| $O_4$ profile | 1% | Wagner et al., 2019 |
| AP & SSA | 3% | Section 6 |
| Strat aerosols | 1% | Section 6.1 |
| albedo | 1% | Section 6 |
| Profile shape | 2% for elevation angles < 4°, negligible for higher elevation angles | Section 6.1 |
| RTM general | 1% | Wagner et al., 2019 |
| total | 4% | |
| | | |
| **$O_4$ VCD** | 2% | This study, section 5, see also Wagner et al., 2019 |







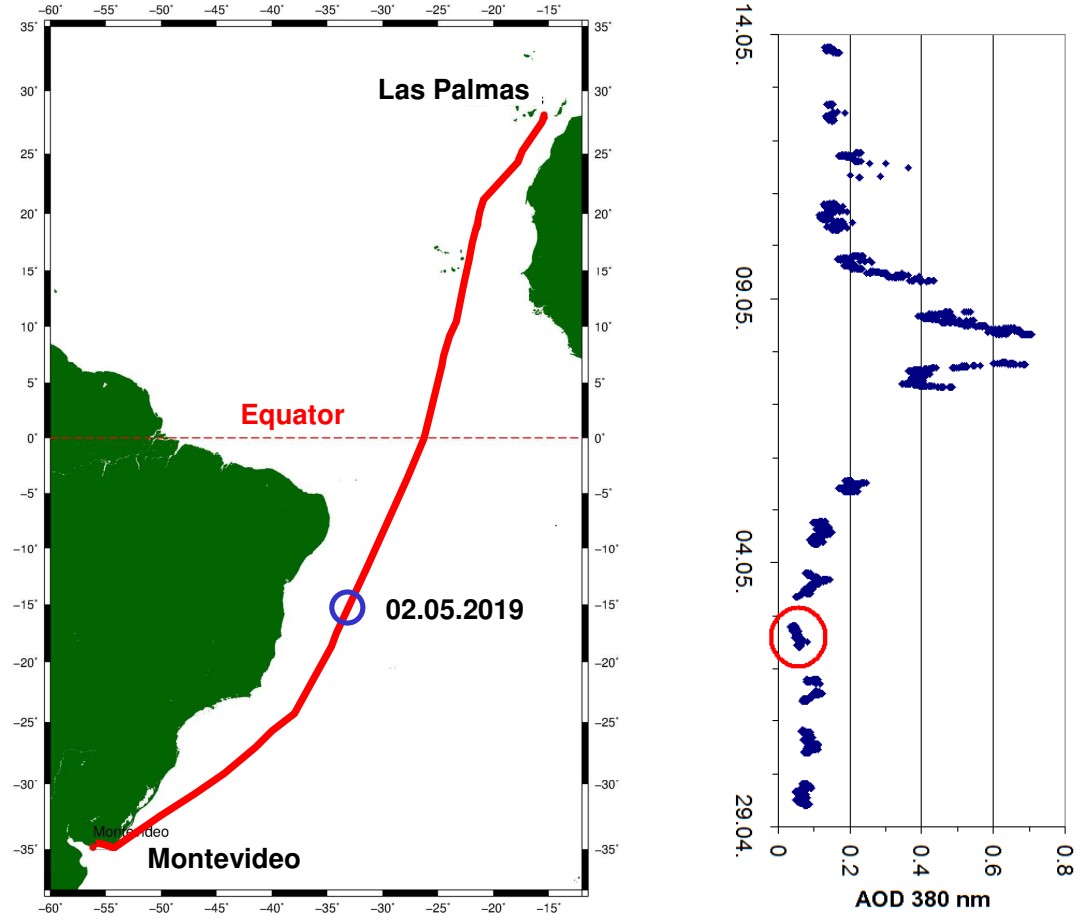


**Fig. 1: Left: Ship route from Montevideo to Las Palmas. The blue circle indicates the location of the measurements used in this study. Right: Aerosol optical depth at 380 nm measured with a hand-held sun photometer.**



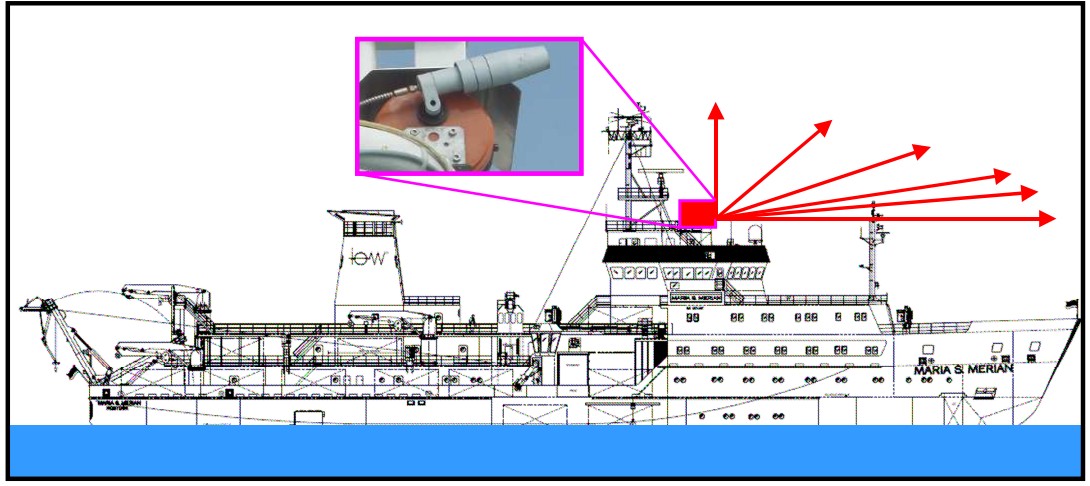

**Fig. 2: The position and viewing direction of the MAX-DOAS instrument on the RV Maria S. Merian during the ship cruise (ship drawing taken from https://briese-research.de/research-department/research-vessels/rv-maria-s-merian).**


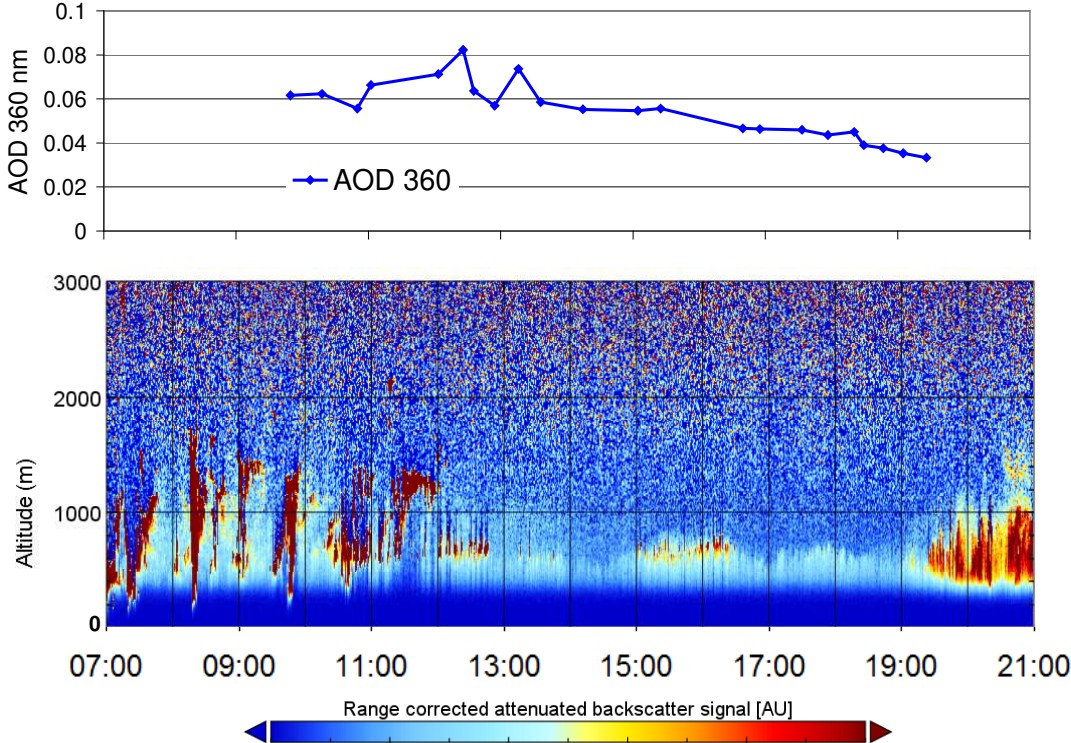

**Fig. 3: Top: AOD at 360 nm measured from the hand-held sun photometer. The data were extrapolated from the**
**measurements at 380 nm using the Angström coefficient calculated from 380 nm and 440 nm. Bottom: Range-corrected ceilometer backscatter profile at 1064 nm.**

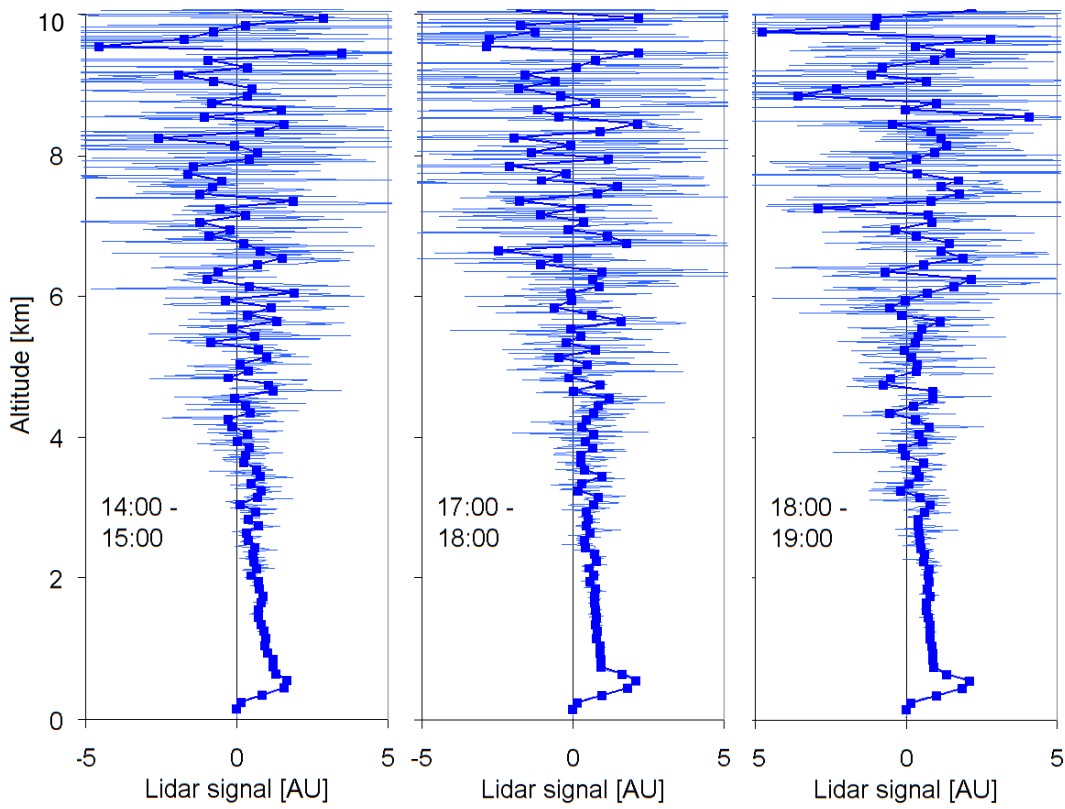


**Fig. 4: Hourly averaged and range corrected ceilometer backscatter profiles for 3 periods on 2 May 2019 without cloud contamination. The thin lines represent the raw data. The dotted lines represent the smoothed profiles (averages over 100 m). The scatter of the range corrected backscatter profiles increases, because the received raw signal scales with the inverse of the square of the distance.**






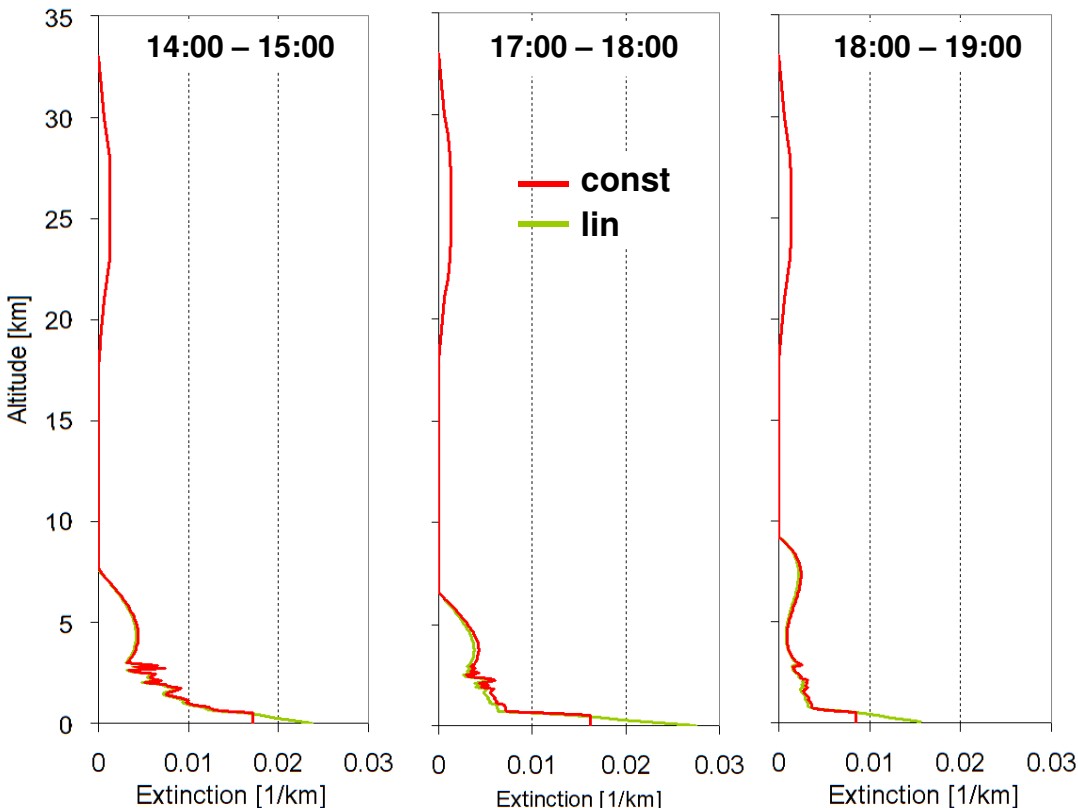

**Fig. 5: Complete aerosol extinction profiles for the three time periods without clouds after all corrections are applied. The green curves represent the profiles with linear extrapolation below 500 m; the red curves represent profiles with constant values below 500 m.**






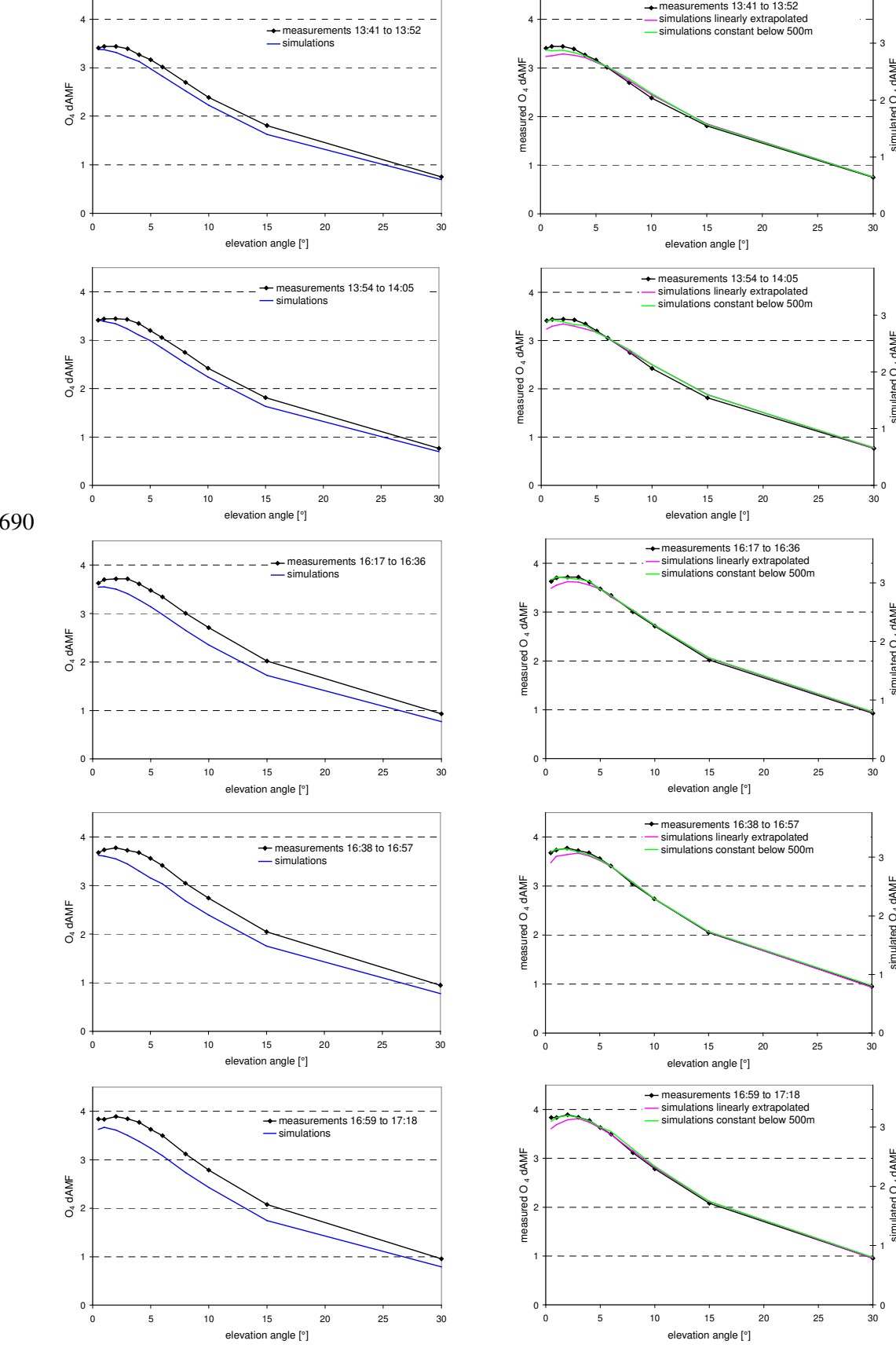

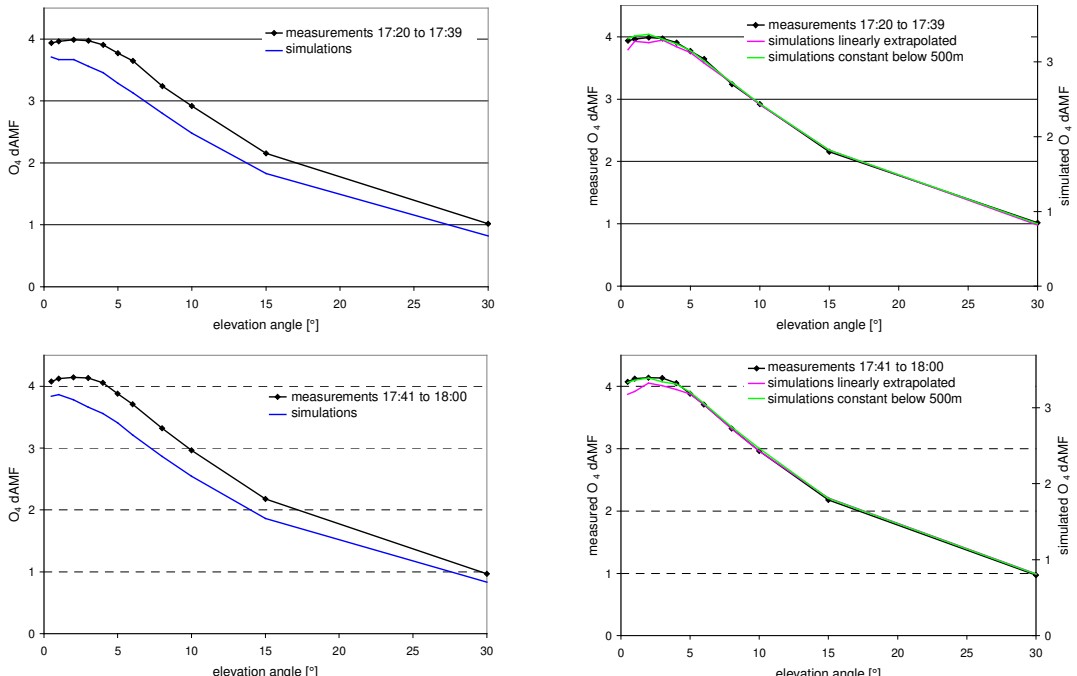


**Fig. 6: Comparison of the measured and simulated O₄ dAMFs for selected elevation sequences without cloud contamination. In the left part, the measured O₄ dAMFs are compared to simulations for a pure Rayleigh atmosphere. In the right part they are compared to simulation results including aerosols (two profiles with either**
**constant or linearly extrapolated aerosol extinction below 500 m). Note that in the right part separate y-axes on the right sides are used for the simulation results. The maxima of the right y-axes are chosen to achieve best agreement between the measured and simulated O₄ dAMFs (see text).**


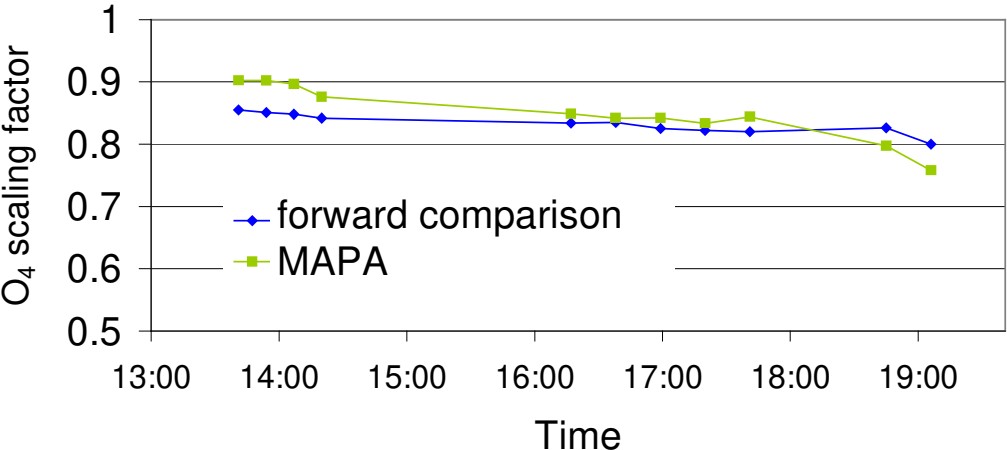


**Fig. 7: Scaling factors derived from the direct comparison between the measured and simulated O₄ dSCDs (blue) and from the MAPA profile inversion (green) for all elevation sequences shown in Fig. 6 and Fig. A12. Many of the measurements of the two last elevation sequences are affected by clouds. Note that for the last elevation sequence, the AOD used in the forward model has large uncertainties, see section 2.2.**



**Appendix A1 Effect of including a H$_2$O cross section or a second O$_4$ cross section on the retrieved O$_4$ dSCDs**

**H$_2$O cross section**

Recent studies found evidence for substantial atmospheric H$_2$O absorptions in measured spectra in the UV (Lampel et al., 2017, Wang et al., 2017, 2020). These absorptions are usually rather small, but especially for measurement conditions with high atmospheric humidity the inclusion of a H$_2$O cross section in the spectral analysis can be useful.

Fig. A1 presents examples of the spectral analysis with either a H$_2$O cross section included or excluded. A clear H$_2$O absorption signal is found around 363 nm. The H$_2$O dSCDs retrieved at 363 nm agree reasonably well (r²=0.63) with those 730 retrieved at 442 nm (see Fig. A2) with a similar slope (2.07) as presented in Lampel et al. (2017) who found a slope of 2.39. If the H$_2$O cross section is not included in the analysis, a systematic structure appears in the residual. Thus, in this study, a water vapour cross section (Polyansky et al. 2018) is included in the spectral analysis. Here it should be noted that compared to other locations, the water vapour absorption during the ship cruise was rather high because most of the measurements were carried out under conditions of high atmospheric temperature and humidity. At other, colder locations, the impact of 735 the H$_2$O absorption might be negligible.

Although the H$_2$O absorption is clearly found in the spectral analysis, the effect of including a H$_2$O cross section or not on the retrieved O$_4$ dSCDs is still rather small. If a H$_2$O cross section is included, the retrieved O$_4$ dSCDs are about 2.5 % larger than without a H$_2$O cross section included.

**O$_4$ cross section at low temperature**

We also investigated the effect of including a second O$_4$ cross section for low temperature (203 K). Before using this cross section in the fit it was orthogonalised with respect to the O$_4$ cross section at 293 K. Including the additional O$_4$ cross section leads to only small changes of the retrieved O$_4$ dSCD of about –1.5 %. Here it is interesting to note that the retrieved O$_4$ 745 dSCDs for the O$_4$ cross section at low temperature were negative and the absolute values much smaller ($<2 \cdot 10^{43}$ molec$^2$/cm$^5$) than those at high temperature ($<6 \cdot 10^{43}$ molec$^2$/cm$^5$). The largest negative O$_4$ dSCDs for the O$_4$ cross section at low temperature were found indicating that the effective atmospheric temperatures decrease with elevation angle (see section 6.2). Also, the correlation between both O$_4$ dSCDs (r²=0.20) is very low. Thus we conclude that the measured spectra do not contain significant O$_4$ absorptions at low temperatures. For the interpretation of this finding it should be noted that low 750 temperatures exist only at higher atmospheric layers. The O$_4$ absorptions at these layers mostly cancel out in the spectral analysis, because the light paths of the measured spectra and the Fraunhofer reference spectra at these layers are very similar. This explains that the retrieved O$_4$ absorptions at cold temperatures are very small. To further confirm this hypothesis, we calculated the effective temperatures for the O$_4$ absorptions on 2 May 2019 (see section 6.2) and found them to be very close to the temperature of the high temperature O$_4$ cross section (293 K). Based on these findings, the O$_4$ results in this study are 755 retrieved without including a second O$_4$ cross section at low temperature. It should, however, be noted that for measurements at other locations and seasons including a second O$_4$ cross section in the spectral analysis might be meaningful.

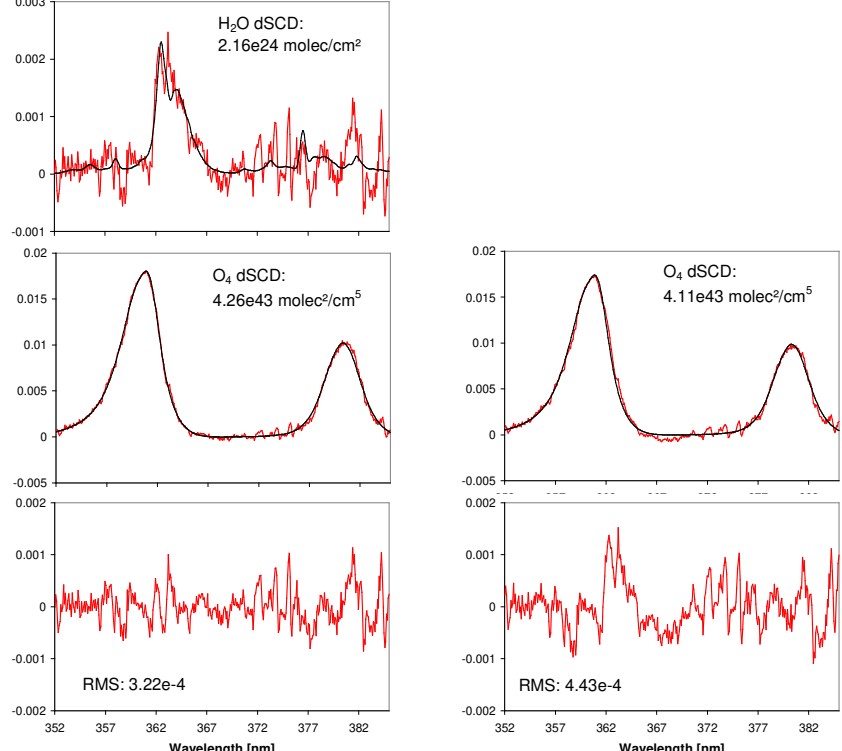


**Fig. A1: Fit results for a spectrum taken on 2 May 2019, 13:14:50, at an elevation angle of 1° (SZA: 33.6°). Left: results if a H$_2$O cross section is included in the spectral analysis; right: results if no H$_2$O cross section is included in the spectral analysis. The black lines represent the fitted cross section, the red lines indicate the residual (bottom) or the residual plus the fitted cross section.**


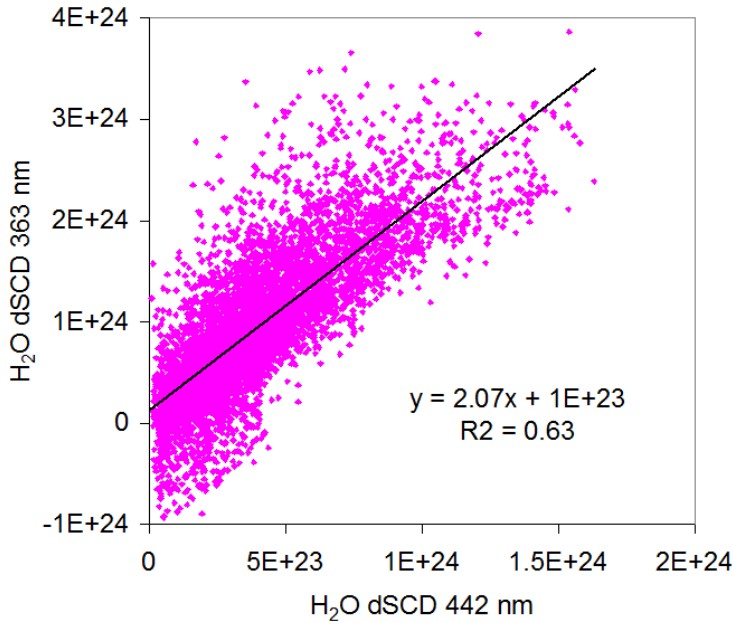

**Fig. A2: Correlation plot of the H$_2$O dSCDs retrieved at 363 nm versus those retrieved at 442 nm for the whole ship**

**cruise. The regression line is fitted assuming that the H$_2$O dSCDs retrieved at 442 nm have no error.**

**Appendix A2 Additional Figures**


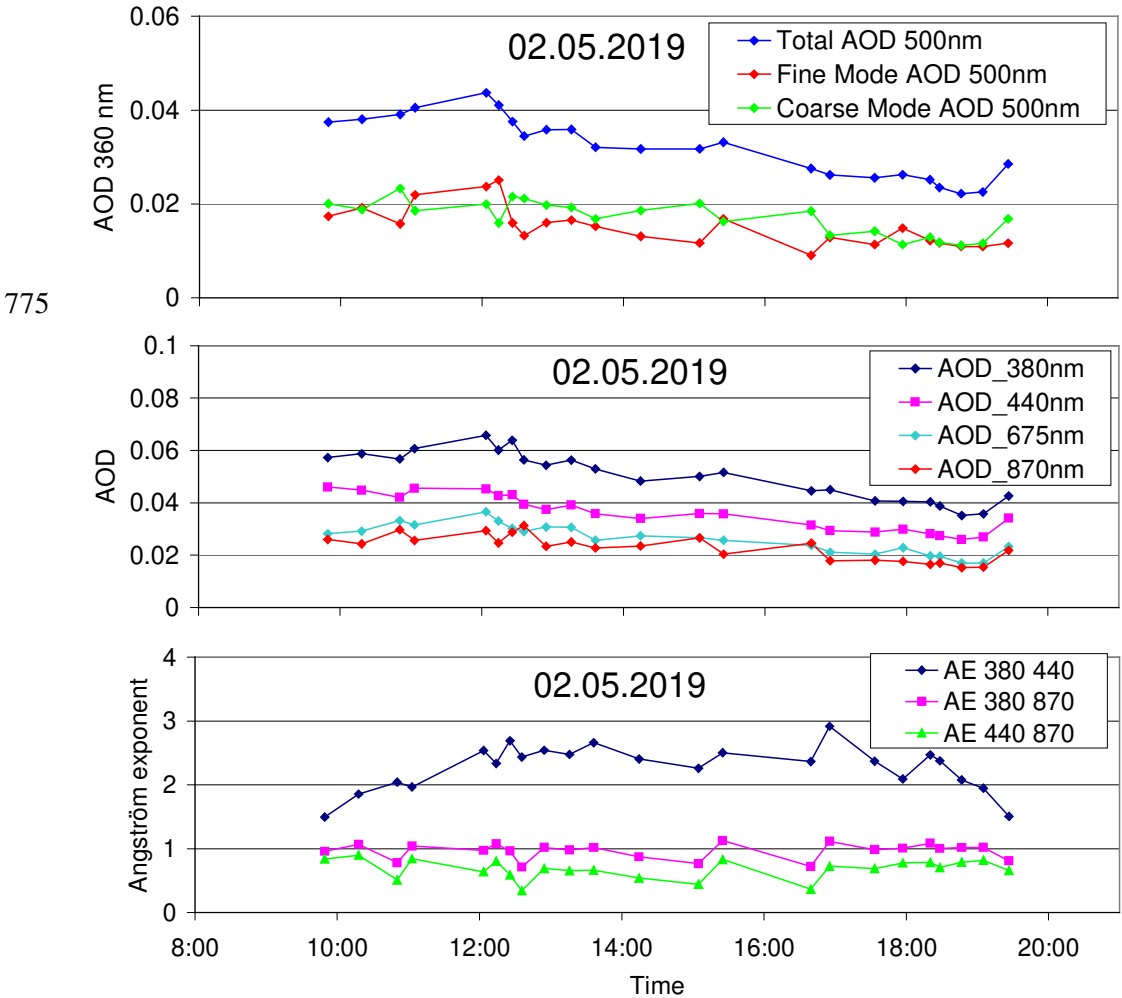

**Fig. A3: Top: AOD at 500 nm attributed to the coarse and fine modes, as well as total AOD. Middle: Total AOD at different wavelengths. Bottom: Angström Exponents for selected wavelength pairs.**


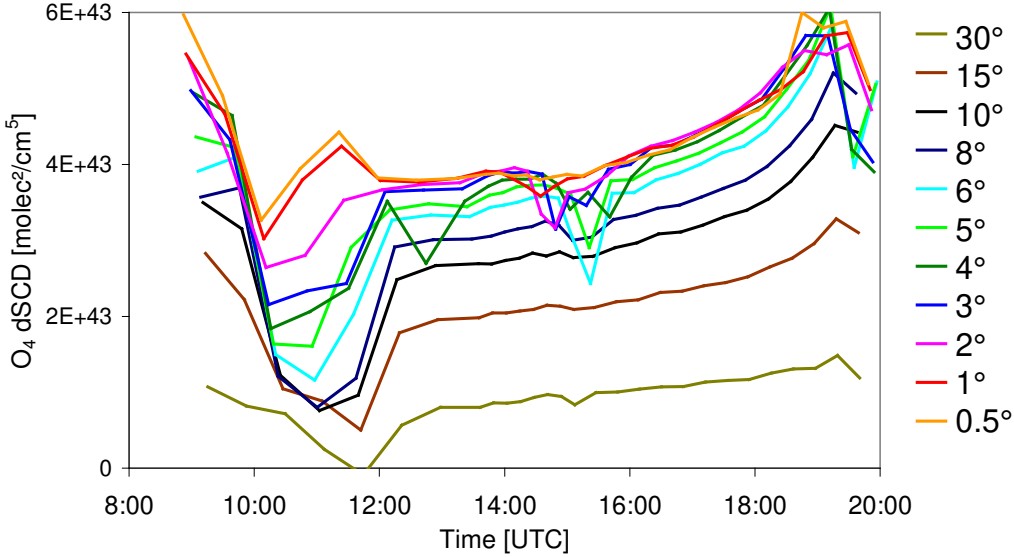

**Fig. A4: Time series of the retrieved O$_4$ dSCD on 2 May 2019 for the different elevation angles. During the afternoon, for most of the time smooth variations are found. However, for some times and elevation angles systematic deviations of the O$_4$ dSCDs occur, which are caused by scattered clouds.**

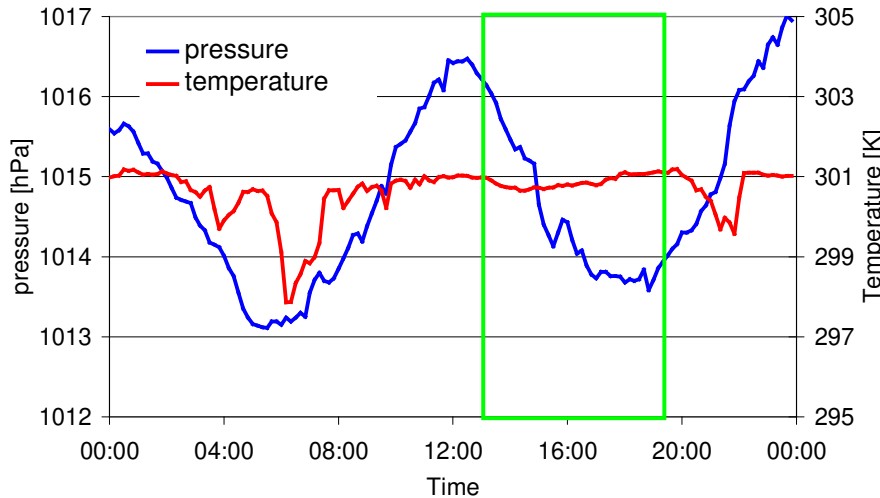

**Fig. A5: Diurnal variation of the surface pressure and temperature from in situ measurements on the ship. The green box indicates the period of the MAX-DOAS measurements used in this study. The corresponding values from the ECMWF model simulations are 1012.8 hPa and 299.8 K, respectively.**

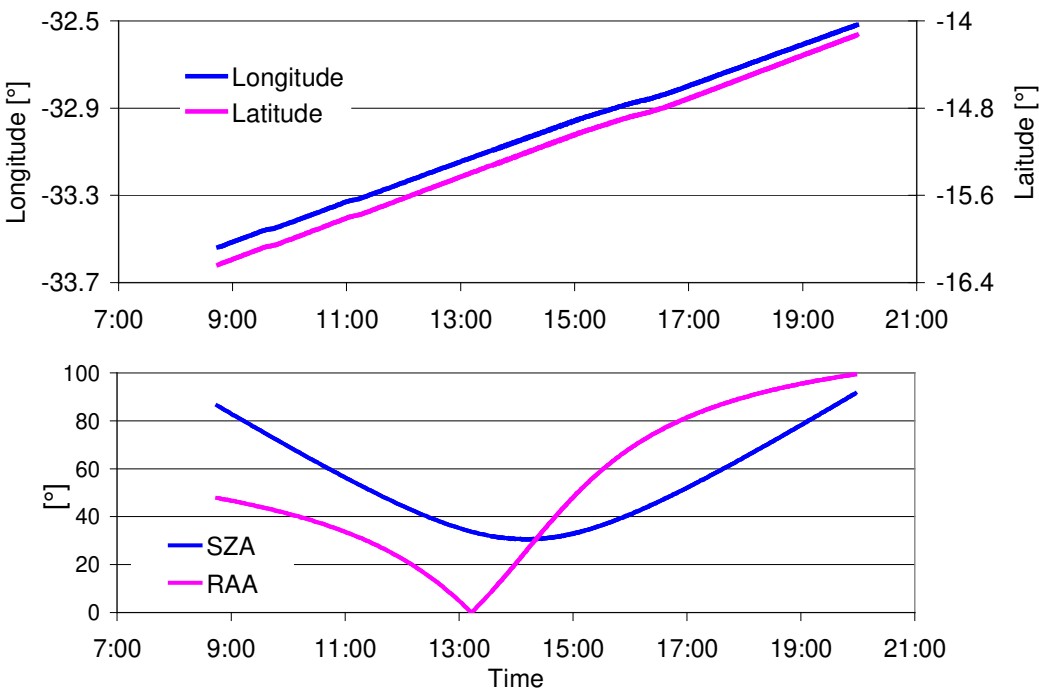

**Fig. A6: Top: Variation of the latitude and longitude of the ship position during 2 May 2019. Bottom: Corresponding variation of the SZA and relative azimuth angle (RAA).**

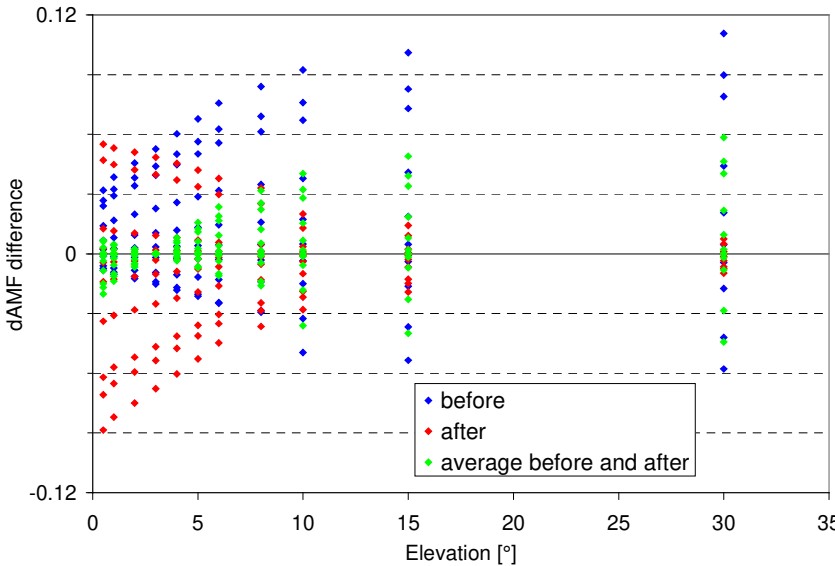

**Fig. A7: Effect of using different Fraunhofer reference spectra for the analysis of individual elevation sequences. Shown are the ratios of the obtained $O_4$ dSCDs for different selections versus those for Fraunhofer reference spectra interpolated between the zenith measurements before and after the selected elevation sequence. Before: zenith measurement before the sequence is used; After: zenith measurement before the sequence is used; Average before and after: the average of the zenith measurements before and after the sequence is used.**

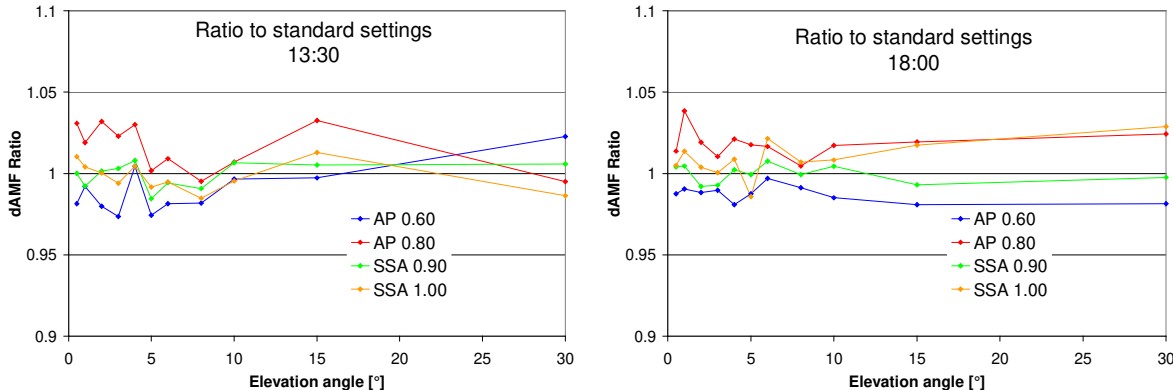

**Fig. A8: Effect of different phase functions and single scattering albedos on the O$_4$ dSCDs. Shown are the ratios for simulations with variations of asymmetry parameter (AP) and single scattering albedo (SSA) versus simulations using the standard settings (AP= 0.68, SSA: 0.95). The results are for SZA of 33.6° and RAA of 0.7° (left, around 13:30) and SZA of 64.5° and RAA of 87.7° (right, around 18:00) on 2 May 2019. The results for other SZA/RAA combinations during the afternoon of 2 May 2019 are similar.**

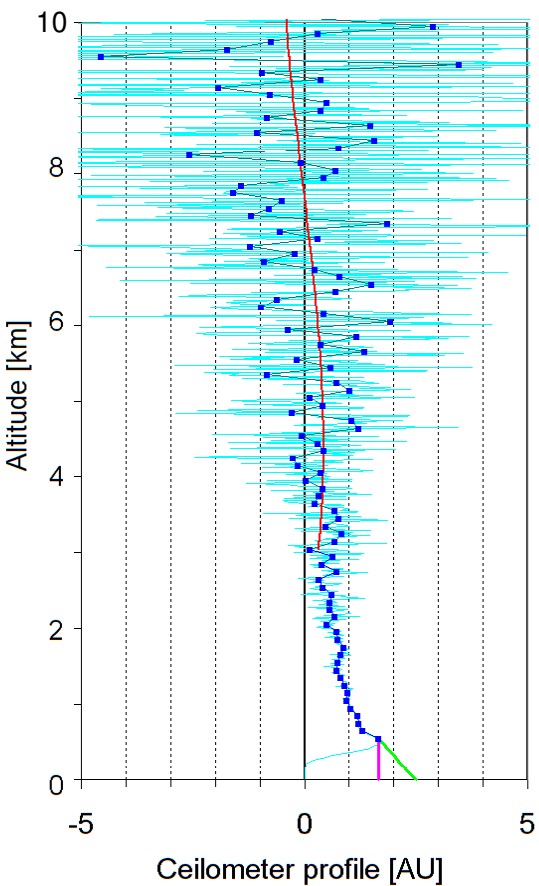

**Fig. A9: The light blue data show the original backscatter profile averaged between 14:00 and 15:00. The blue dots show the smoothed (with a 100m kernel) profile, which are used between 500 m and 3 km. Below 500m either constant or linearly extrapolated data (see text) are used. Between 3 km and 10 km a third order polynomial is fitted to the raw data. The polynomial values are used between 3 km and the altitude at which they become negative. Above, the values are set to zero.**

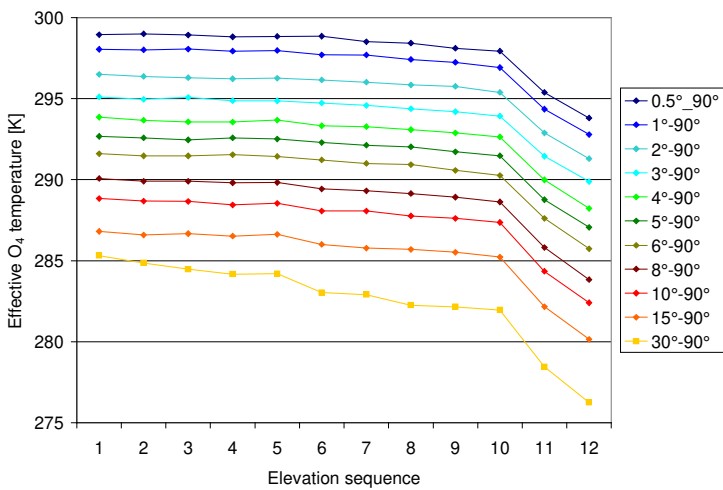

**Fig. A10: Effective temperatures calculated for the individual measurements according to equation 2.**


**Fig. A11: Effect of the temperature correction for two selected elevation sequences.**




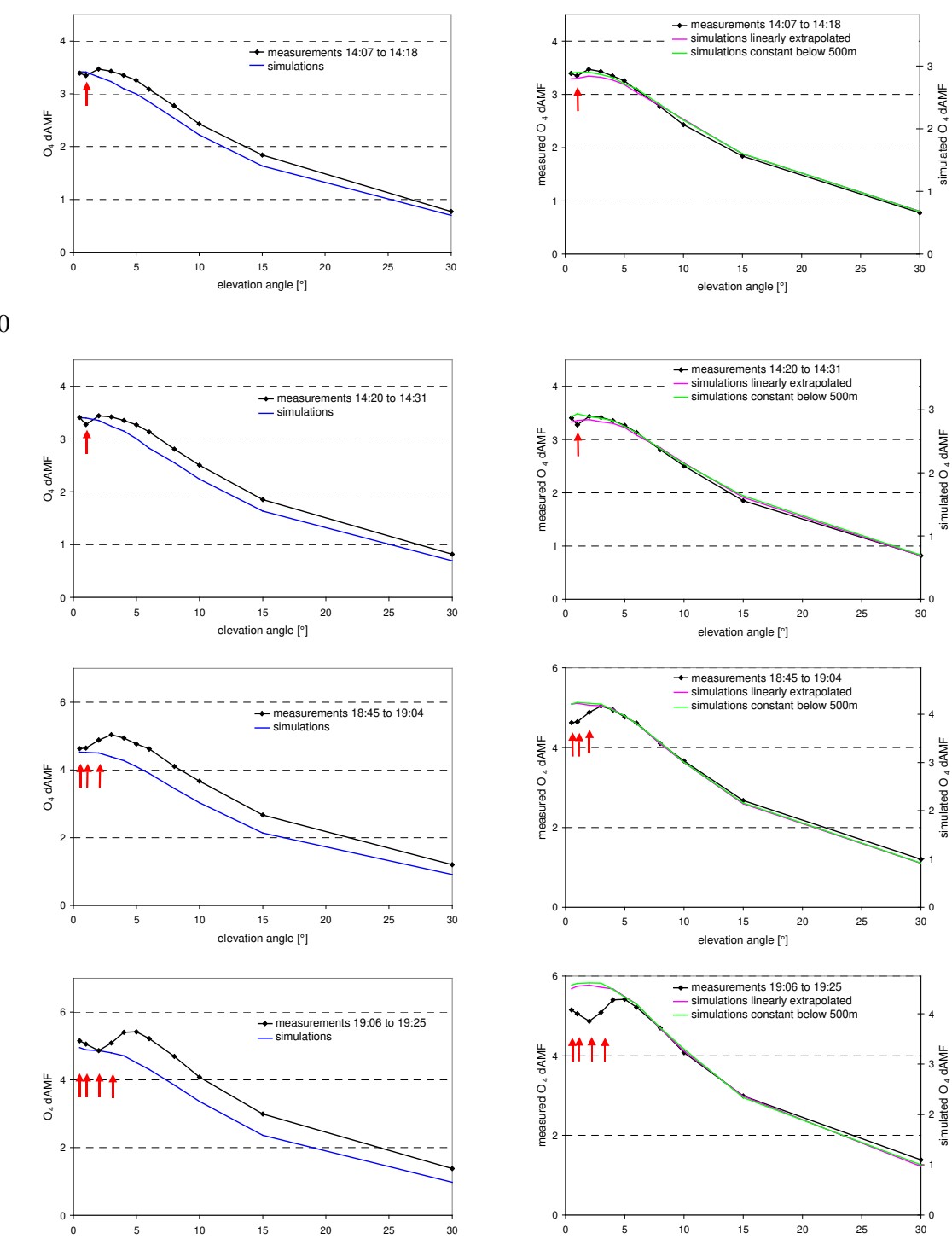


**Fig A12: Comparison of the measured and simulated O₄ dAMFs for five elevation sequences with few cloud-contaminated measurements (indicated by the red arrows). In the left part, the measured O₄ dAMFs are compared to simulations for a pure Rayleigh atmosphere. In the right part they are compared to simulation results including aerosols (two profiles with either constant or linearly extrapolated aerosol extinction below 500 m). Note that in the right part separate y-axes on the right sides are used for the simulation results. The maxima of the right y-axes are chosen to achieve best agreement between the measured and simulated O₄ dAMFs (see text). Note that for the last elevation sequence (19:06 – 19:25), the AOD used in the forward model has large uncertainties, see section 2.2.**



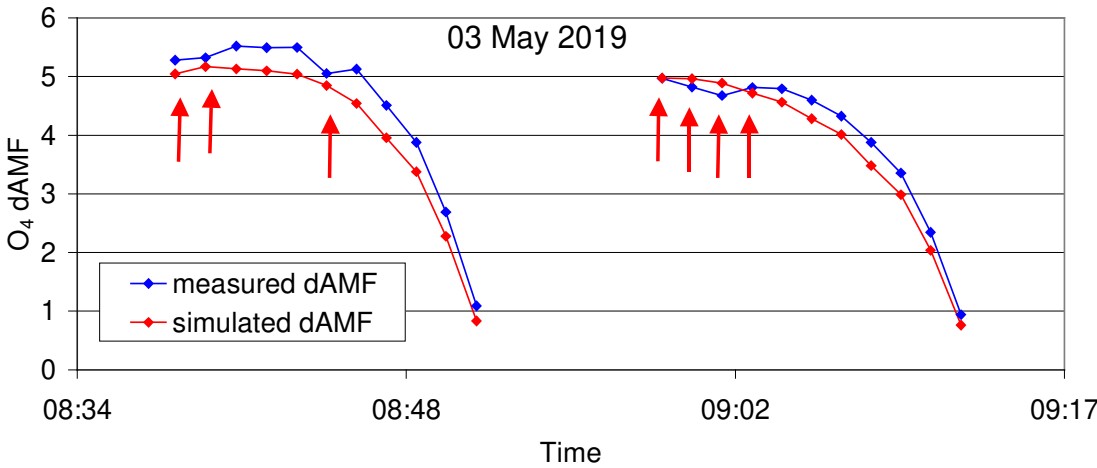


**Fig A13: Comparison of the measured and simulated O$_4$ dAMFs for two elevation sequences on 05 March 2019. For the first elevation sequence, the AOD was <0.05 at 360 nm. During the second elevation sequence it already increased to 0.06. The radiative transfer simulations were made for an aerosol-free atmosphere. The red arrows indicate cloud-contaminated measurements.**






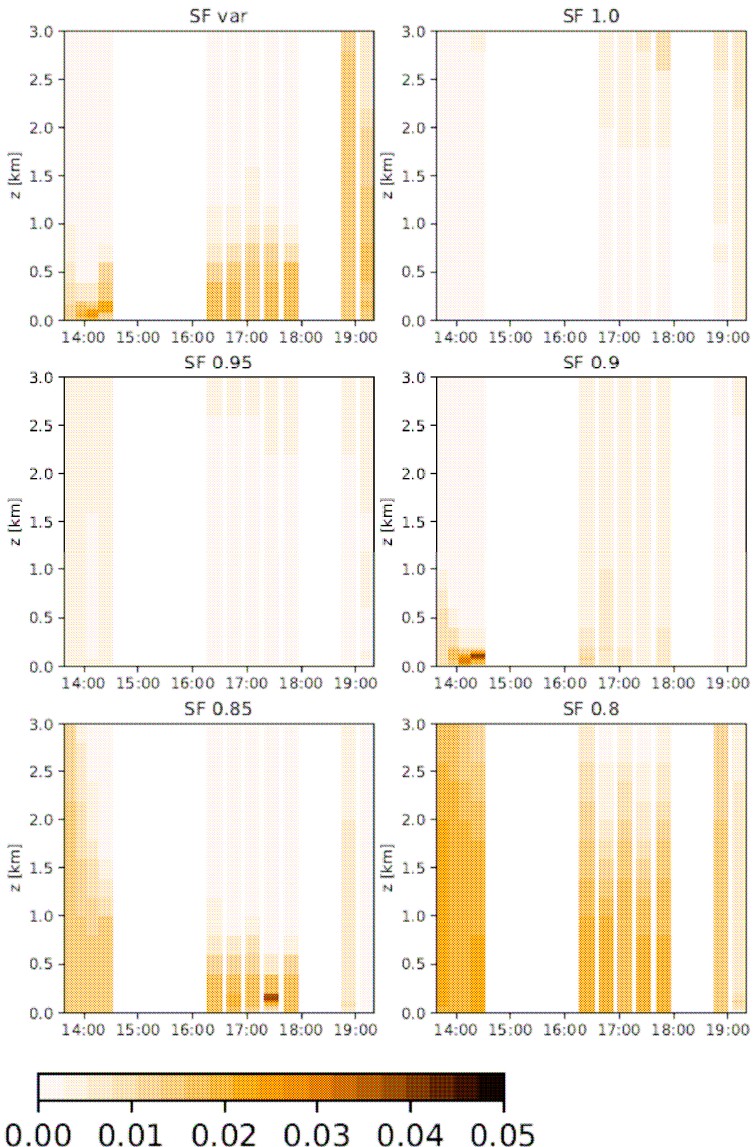

 **Fig A14: Extinction profiles retrieved with MAPA for the selected elevation sequences for different scaling factors. Only profiles for inversions with ‚valid' or ‚warning' flags are shown.**


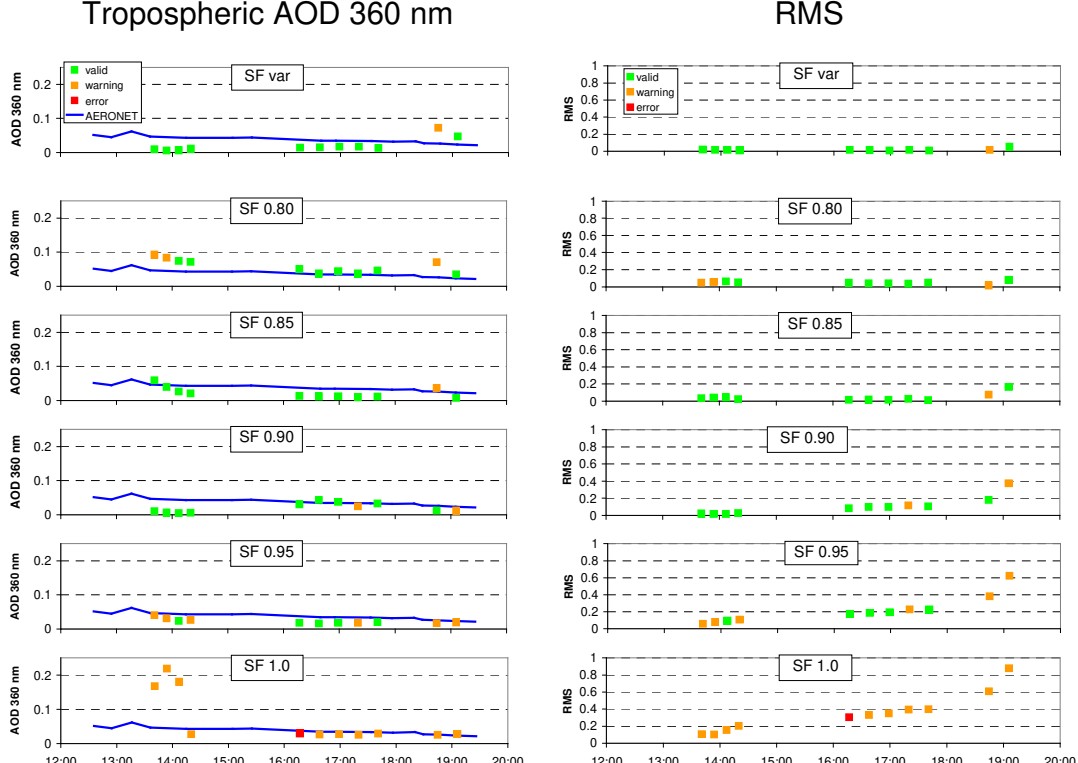

**Fig A15: Right:** Comparison of the retrieved AOD for the MAPA profile inversions with different scaling factors (squares) and the (tropospheric) AOD observed by the sun photometer (blue lines). **Right:** RMS between the measured and fitted $O_4$ dAMFs. The colours indicate the quality flags for the individual profile inversions.