# Peer review of "Quantitative comparison of measured and simulated $O_4$ absorptions for one day with extremely low aerosol load over the tropical Atlantic"

_Atmospheric Measurement Techniques, 2020_

## Referee Comment (RC1) · Anonymous Referee #1 · 11 Feb 2021

This manuscript addresses an inconsistency persistently reported in several past studies (some of them by the same authors) between observations of the O4 absorption in atmospheric spectra and radiative transport simulations attempting to reproduce these observations. This inconsistency represents a major issue for the interpretation of MAX-DOAS measurements of aerosol properties, which are based on O4 slant column measurements. Results from past studies indicated that, to reconcile observations with simulations, it is often (but not always) necessary to apply a scaling factor of typically -20% to observations. The reason why this correction is needed remains unknown, but some authors suggested that it might be related to uncertainties in the knowledge of aerosol properties in the atmosphere, which can possibly affect the light path of the

solar radiation in a complex way. In the present study, the authors try to overcome this difficulty by concentrating their analysis on observations performed under very low AOD conditions, therefore minimizing uncertainties due to aerosols. Even in such particular conditions, they find that simulations underestimate measurements by about 20%, which confirms that a fundamental inconsistency - not related to aerosols - exists between observations and simulations. Although the study is limited in coverage (only one day of measurements is presented), the proposed case is fully pertinent as it suggests that at least for the conditions of the study inconsistencies cannot be resolved by uncertainties in aerosol properties. The mystery remains however unresolved, since no valid explanation can be proposed. The suggestion that systematic errors on the O4 spectroscopy could be an explanation is on the one hand in contradiction with known uncertainties on laboratory measurements, and on the other hand also in contradiction with published results indicating that a scaling factor is not always required to bring measurements and simulations in good agreement. It would of course be interesting to multiply measurements in similarly low AOD conditions but this is clearly beyond the scope of the paper. From an editorial point of view, the manuscript is concise, well written and well organized. I therefore recommend its publication in AMT, after attention to the few comments listed below.

Specific comments

Pg. 5, l. 3: please justify the use of 0.05 as appropriate value for the albedo of the sea at UV wavelengths. A reference would be enough here. Also indicate at which wavelength the radiative transfer calculations have been computed.

Pg. 5, l. 205: I have the impression that the use of lidar backscatter ratio profiles as a proxy for aerosol extinction profiles involves more assumptions than stated here. E.g., one also has to assume that the aerosol phase function does not vary much with altitude, and maybe more important that the backscatter profile shape measured at 1000 nm is also valid at 360nm. But despite all uncertainties, I agree that using ceilometer profiles makes sense in the absence of real extinction values. Maybe the

system could be improved by adding a device to measure aerosol surface extinctions (if possible).

Pg. 5, l. 221: add a reference to justify the Angström exponent of 2 used for the conversion of the stratospheric AOD (unless this would be documented in Thomason et al., 2018)

Pg. 7, sect. 7.2: considering the very low aerosol content, and the comparatively large uncertainty of the assumed stratospheric AOD (basically a climatological value at 525 nm converted to 360 nm using a not well established Angstrom exponent), I think that the AOD values retrieved by MAPA are highly uncertain. The fact that the retrieved scaling factor matches the values empirically derived in the previous section is not really surprising, since this scaling is already necessary to bring clear-sky simulations in agreement with observations. Inspecting more closely Fig. A13, it seems that the retrieved AOD values are very unstable. Comparing e.g. results derived using SF=0.8 and SF=0.85, we see that AOD values differ quite substantially although RMS values are similar. I am not really convinced that MAPA inversions add a lot of information in the study. At least they are not inconsistent. Something that would be very interesting would be to test whether the discrepancy depends on the O4 wavelength used for the retrieval. Unfortunately, this is not possible using the current setup due to the limited spectral range of the spectrometer, but it should be considered for future studies. Finally, one may also wonder whether this particular day was really the only clean day (during the ship cruise) allowing for a comparison of measured and simulated O4 slant columns. If other similarly clean days were encountered, it would be nice to know whether similar inconsistencies were found.

Spelling, typos:

Pg. 1, l. 21: remove 'variation'

Pg. 1, l. 33: remove 'mainly'

Pg. 3, l. 101: add 'at' between 'are not' and 'the identical location'

Pg. 4, l. 151: add 'dry air' between 'For the' and 'mixing ratio of O2'

---

## Referee Comment (RC2) · Anonymous Referee #2 · 2 Mar 2021

**General comments**

In recent years, more and more authors have indicated the need for scaling factors in order to improve the agreement of measured and simulated $O_4$ dSCD/dAMF. In the previous publication by Wagner et al. 2019, various factors were investigated to determine the possible cause of this disagreement. One of the key remarks made by reviewers and the community was that the uncertainty of aerosol information and its impact on the oxygen dimer could not be ruled out as a possible cause of disagreement. In this novel study, Wagner et al. examine the difference of measured and simulated

$O_4$ dAMF for low aerosol loads measured during a ship cruise in the Atlantic in 2019. The authors claim that due to the low aerosol load possible aerosol uncertainties can be neglected and that the underlying differences must have another, as yet unknown reason.

The document is well written and structured and the analyses have been carried out thoroughly and consistently. However, I recommend publishing it after making some minor changes listed below.

1. Please add a table including all uncertainties described in the document (e.g. pressure/temperature changes, aerosol parameterization, effective temperature, ...)

2. It would be interesting to have a time series of $O_4$ dSCD/dAMF RMS values (similar to A13) for the data shown in Fig.6 and A11. I would expect a clear trend in the RMS differences over the day maybe similar to the one you showed for AOD and scaling factor? How is the correlation of these RMS values and the retrieved/measured AOD?

3. You mentioned that sun photometer measurements allow to differ between the aerosol particle size. Please show the contribution of differently sized aerosol particles to the total AOD over the day as well as all AODs and corresponding Angström exponents.

4. Furthermore, I was wondering why you only showed results for one day? The AODs for the following days are also rather small. Do these days support your findings?

**Specific comments**

**P1, L22, 25:** Please add a selection of corresponding references to the sentences starting with (L22) "In recent years,..." and (L25) "Several studies found that a scaling factor...".

**P2, Sec 2.2 and Fig.3:** Since AODs at other wavelengths are available, please add them to Fig. 3.

**P3, L87:** "(with..." $\Rightarrow$ "(which...?

**Fig. A1:** Your fit uses the wavelength range 352 - 387nm but Fig. A1 shows only wavelengths up to $\sim$384nm. Please change the x-Axis according to the applied fitting window. Furthermore, I was wondering about the shown residual. It appears to me that there are still some residual structures left. Especially three peaks around 372-376 nm look familiar and could be attributed to Fraunhofer-Lines. Could this be somehow related to your Ring-treatment or do you have another explanation?

**P5, L177:** Why was the albedo set to 0.05? Please add a references here. As far as I know, we can expect a small dependence on SZA. How large is the impact on $O_4$ when changing the albedo according to possible values?

**P6, L215:** "the the", please remove either first or second "the".

**P6, L216:** You wrote that Fig. A8 includes constant and linearly extrapolated values for lower altitudes but the greenish line does not look like a linear extrapolation to me. Why is that?

**P6, L221:** Why is the Angström exponent "assumed" to be 2 when you have AODs at several wavelengths available to calculate more accurate values?

**P6, L224:** decsribed $\Rightarrow$ described

**P7, L263:** "smaller than" $\Rightarrow$ "larger than"

**P7, L276:** "bebetween" $\Rightarrow$ "be between"

**Fig. A7** Could you please add a similar figure for the geometry with the smallest RAA to better assess the impact of AP and SSA variations throughout the day?

**References**

Wagner, T., Beirle, S., Benavent, N., Bösch, T., Chan, K. L., Donner, S., Dörner, S., Fayt, C., Frieß, U., García-Nieto, D., Gielen, C., González-Bartolome, D., Gomez, L., Hendrick, F., Henzing, B., Jin, J. L., Lampel, J., Ma, J., Mies, K., Navarro, M., Peters, E., Pinardi, G., Puentedura, O., PuÄůÄńte, J., Remmers, J., Richter, A., Saiz-Lopez, A., Shaiganfar, R., Sihler, H., Van Roozendael, M., Wang, Y., and Yela, M.: Is a scaling factor required to obtain closure between measured and modelled atmospheric O4 absorptions? An assessment of uncertainties of measurements and radiative transfer simulations for 2 selected days during the MAD-CAT campaign, Atmos. Meas. Tech., 12, 2745–2817, https://doi.org/10.5194/amt-12-2745- 405 2019, 2019
* * *

---

## Referee Comment (RC3) · Anonymous Referee #3 · 3 Mar 2021

The present manuscript addresses the issue of the difference between measured and simulated O4 dSCDS. Many studies over the last year used correction factors on the measured O4 dSCDs to achieve a better agreement without finding the physical explanation of these factors. Other studies support that the use of correction factor is not necessary. In previous studies, one possible explanation of this inconsistency was the uncertainties of aerosol information. It is very interesting that in this manuscript, this uncertainty is neglected because of the use of one day of measurements with very low AOD values.

I recommend the publication of the present manuscript. The content is clear, well

explained and the manuscript falls into the scope of AMT. Please consider some minor comments:

1. In Figure 1, I see that other days (or at least time windows during some days) have very small AOD values. Why these days are not included in your results? Would it be possible to include them and see if the results agree with the main findings of your study?

2. The uncertainties that are described in the text are very important for your findings. That would be very useful if you could create a table of uncertainties.

Specific comments:

1. P.1, Line 21 : "aside from" instead of "aside"

2. P.1, Line 22 : "e.g.," instead of "e.g."

3. P. 1, Line 32 : "In this study," instead of "In this study"

4. P. 1, Line 25 : Please add some studies that used a scaling factor

5. P. 4, Line 156 : Can you provide a possible explanation for this difference between ECMWF and in-situ measurements?

6. P. 5, Line 175 : "simulations," instead of "simulations"

7. P. 5, Line 208, 210, Figure 4 : Is there any explanation why the raw data vary more with altitude? Is it valid to use the data above 3 km?

8. Figure A8 : y axis varies from 0 km to 10 km and not from 0 km to 0 km. Please correct

9. P. 6, Line 221 : why the Angstrom coefficient is assumed equal to 2?

10. P. 7, Line 276 : "be between" instead of "bebetween"

---

## Author Comment (AC1) · 29 Mar 2021

**Reply to Referee #1**

First of all we want to thank this reviewer for the positive assessment and constructive comments.

We addressed these comments as explained in detail below.

This manuscript addresses an inconsistency persistently reported in several past studies (some of them by the same authors) between observations of the O4 absorption in atmospheric spectra and radiative transport simulations attempting to reproduce these observations. This inconsistency represents a major issue for the interpretation of MAX-DOAS measurements of aerosol properties, which are based on O4 slant column measurements. Results from past studies indicated that, to reconcile observations with simulations, it is often (but not always) necessary to apply a scaling factor of typically -20% to observations. The reason why this correction is needed remains unknown, but some authors suggested that it might be related to uncertainties in the knowledge of aerosol properties in the atmosphere, which can possibly affect the light path of the solar radiation in a complex way. In the present study, the authors try to overcome this difficulty by concentrating their analysis on observations performed under very low AOD conditions, therefore minimizing uncertainties due to aerosols. Even in such particular conditions, they find that simulations underestimate measurements by about 20%, which confirms that a fundamental inconsistency - not related to aerosols - exists between observations and simulations. Although the study is limited in coverage (only one day of measurements is presented), the proposed case is fully pertinent as it suggests that at least for the conditions of the study inconsistencies cannot be resolved by uncertainties in aerosol properties. The mystery remains however unresolved, since no valid explanation can be proposed. The suggestion that systematic errors on the O4 spectroscopy could be an explanation is on the one hand in contradiction with known uncertainties on laboratory measurements, and on the other hand also in contradiction with published results indicating that a scaling factor is not always required to bring measurements and simulations in good agreement. It would of course be interesting to multiply measurements in similarly low AOD conditions but this is clearly beyond the scope of the paper. From an editorial point of view, the manuscript is concise, well written and well organized. I therefore recommend its publication in AMT, after attention to the few comments listed below.

**Many thanks for this positive assessment.**

**Specific comments**

Pg. 5, l. 3: please justify the use of 0.05 as appropriate value for the albedo of the sea at UV wavelengths. A reference would be enough here. Also indicate at which wavelength the radiative transfer calculations have been computed.

We chose the value of 5% to be consistent with the MAPA inversions, and because it is appropriate for many parts of the global ocean. However, by having a closer look at maps of albedo (Kleipool et al., 2008) and chlorophyll content (e.g. from the NASA Earth Observatory: https://earthobservatory.nasa.gov/globalmaps/MY1DMM\_CHLORA), we found that at the specific location of the measurements, very clear waters exist, for which the surface albedo is typically higher (about 7 to 8%). The presence of very clear waters is also supported by the in situ chlorophyll measurements made aboard the ship.

We therefore made additional radiative transfer simulations using a surface albedo of 8%. We found that the obtained  $O_4$  dAMFs were almost identical with those obtained for 5% surface albedo (differences <1%). The reason for the good agreement is that the effect of the surface albedo is similar for the  $O_4$  AMFs for different elevation angles. Thus the effect of varying surface albedo almost cancels out. We added this information to section 6.

Pg. 5, l. 205: I have the impression that the use of lidar backscatter ratio profiles as a proxy for aerosol extinction profiles involves more assumptions than stated here. E.g., one also has to assume that the aerosol phase function does not vary much with altitude, and maybe more important that the backscatter profile shape measured at 1000 nm is also valid at 360nm. But despite all uncertainties, I agree that using ceilometer profiles makes sense in the absence of real extinction values. Maybe the system could be improved by adding a device to measure aerosol surface extinctions (if possible).

Unfortunately, no measurements of aerosol surface extinction are available.

With respect to the representativeness of the measurements at 1064 nm for the MAX-DOAS measurements, we agree with the reviewer that the aerosol properties can change with altitude, and thus the relative profile shape of aerosol extinction at 360 nm might differ from that at 1064 nm.

In order to estimate the effect of the varying aerosol profiles at both wavelengths, we performed additional radiative transfer simulations using modified tropospheric aerosol profiles. The aerosol extinction in the lowest 1000 m of the extracted profiles was changed by +/-20% and the free tropospheric part above was adjusted to keep the total AOD unchanged. The resulting O4 dAMFs were almost unchanged for elevation angles >4°. For lower elevation angles, the changes were found to be +/-2%. We added this information to section 6.1.

Pg. 5, l. 221: add a reference to justify the Angström exponent of 2 used for the conversion of the stratospheric AOD (unless this would be documented in Thomason et al., 2018)

We took the value of 2 from existing publications (e.g. Malinina et al., 2019). However, most provided values are representative for larger wavelengths (typically 525 nm or larger). To estimate the uncertainties of the simulated O4 dSCDs related to the uncertainty of the Angström exponent, we performed additional radiative transfer simulations assuming a stratospheric AOD of 0.008 (corresponding to an Angström exponent of 1). We found that the O4 dSCDs differ from those for a stratospheric AOD of 0.012 by less than 1%. We added this information to section 6.1.

Pg. 7, sect. 7.2: considering the very low aerosol content, and the comparatively large uncertainty of the assumed stratospheric AOD (basically a climatological value at 525 nm converted to 360 nm using a not well established Angstrom exponent), I think that the AOD values retrieved by MAPA are highly uncertain. The fact that the retrieved scaling factor matches the values empirically derived in the previous section is not really surprising, since this scaling is already necessary to bring clear-sky simulations

in agreement with observations. Inspecting more closely Fig. A13, it seems that the retrieved AOD values are very unstable. Comparing e.g. results derived using SF=0.8 and SF=0.85, we see that AOD values differ quite substantially although RMS values are similar. I am not really convinced that MAPA inversions add a lot of information in the study. At least they are not inconsistent. Something that would be very interesting would be to test whether the discrepancy depends on the O4 wavelength used for the retrieval. Unfortunately, this is not possible using the current setup due to the limited spectral range of the spectrometer, but it should be considered for future studies. Finally, one may also wonder whether this particular day was really the only clean day (during the ship cruise) allowing for a comparison of measured and simulated O4 slant columns. If other similarly clean days were encountered, it would be nice to know whether similar inconsistencies were found.

We fully agree with the reviewer that the aerosol results from MAPA have large uncertainties.

In the original manuscript, we already wrote: ,However, here it should be noted that for these low aerosol extinctions, the information content of the measurements is probably too low to constrain the aerosol extinction profiles, especially for high altitudes.'

In the revised version we modified this sentence to: ,However, here it should be noted that for these low aerosol extinctions, the information content of the measurements is probably too low to constrain the aerosol extinction profiles, especially for high altitudes. Thus also the retrieved AOD values are very unstable (see Fig. A15). Nevertheless, rather clear results for the scaling factor are found:'

For the other two points, we made the following changes:

**-other wavelengths:**

We added the following sentence at the end of the conclusions: ,We recommend that similar studies under extremely low aerosol load should be made at different locations and seasons. Also  $O_4$  absorptions at different wavelengths should be investigated.'

**-other measurement days:**

The extremely low AODs only occurred on the selected day. Only at the beginning of the following day, still low AODs were measured (

Fig. A13 Comparison of the measured and simulated  $O_4$  dAMFs for two elevation sequences on 05 March 2019. For the first elevation sequence, the AOD was <0.05 at 360 nm. During the second elevation sequence it already increased to 0.06. Note that the radiative transfer simulations were made for an aerosol-free atmosphere.

Like on 02 May 2019, the simulated  $O_4$  dAMFs (for aerosol-free atmosphere) are smaller than the measurements (for cloud-free observations).

We added the following information to section 7.1.:

,It should be noted that during the entire ship cruise, only during the beginning of 3 May 2019, similarly low (but still larger) AOD were measured as on 2 May 2019. We compared the measured  $O_4$  dAMFs for the first two elevation sequences on 3 May with radiative transfer simulations. For that comparison we only made simulations for an aerosol-free atmosphere in order to to limit the effort (and also because of the rapid temporal variation of the AOD during that time period). The results (see Fig. A13) are similar to those on 2 May 2019: Except for the cloud contaminated measurements, the simulations are smaller than the measurements.'

Spelling, typos:

Pg. 1, l. 21: remove 'variation' Corrected

Pg. 1, l. 33: remove 'mainly' Corrected

Pg. 3, l. 101: add 'at' between 'are not' and 'the identical location' Corrected

Pg. 4, l. 151: add 'dry air' between 'For the' and 'mixing ratio of O2' Corrected

**References**

Kleipool, Q., Dobber, M., de Haan, J., and Levelt, P., Earth surface reflectance climatology from 3 years of OMI data, J. Geophys. Res.-Atmos., 113, D18308, https://doi.org/10.1029/2008JD010290, 2008.

Malinina, E., Rozanov, A., Rieger, L., Bourassa, A., Bovensmann, H., Burrows, J. P., and Degenstein, D.: Stratospheric aerosol characteristics from space-borne observations: extinction coefficient and Ångström exponent, Atmos. Meas. Tech., 12, 3485–3502, https://doi.org/10.5194/amt-12-3485-2019, 2019.

---

## Author Comment (AC2) · 29 Mar 2021

**Reply to Referee #2**

First of all we want to thank this reviewer for the positive assessment and constructive comments.
We addressed these comments as explained in detail below.

General comments
In recent years, more and more authors have indicated the need for scaling factors in order to improve the agreement of measured and simulated O4 dSCD/dAMF. In the previous publication by Wagner et al. 2019, various factors were investigated to determine the possible cause of this disagreement. One of the key remarks made by reviewers and the community was that the uncertainty of aerosol information and ist impact on the oxygen dimer could not be ruled out as a possible cause of disagreement. In this novel study, Wagner et al. examine the difference of measured and simulated O4 dAMF for low aerosol loads measured during a ship cruise in the Atlantic in 2019. The authors claim that due to the low aerosol load possible aerosol uncertainties can be neglected and that the underlying differences must have another, as yet unknown reason.
The document is well written and structured and the analyses have been carried out thoroughly and consistently. However, I recommend publishing it after making some minor changes listed below.

We thank the reviewer for the positive assessment.

1. Please add a table including all uncertainties described in the document (e.g. pressure/temperature changes, aerosol parameterization, effective temperature, ...)

Many thanks for this good suggestion! We added such a table (new table 3) to the paper:

Table 3 Uncertainties related to the different analysis steps

| Spectral analysis | | |
|---|---|---|
| **Effect** | **Magnitude** | **Reference** |
| Spectral fit | 1 - 4% | Result of spectral fit |
| Temperature dependence | 1.5% | Wagner et al., 2019 |
| Fit paramaters | 3.5% | Appendix A1, and Wagner et al., 2019 |
| Total | 4 – 5.5% | |
| | | |
| **RTM without aerosols** | | |
| $O_4$ profile | 1% | Wagner et al., 2019 |
| albedo | 1% | Section 6 |
| RTM general | 1% | Wagner et al., 2019 |
| total | 2% | |
| | | |
| **RTM with aerosols** | | |
| $O_4$ profile | 1% | Wagner et al., 2019 |
| AP & SSA | 3% | Section 6 |
| Strat aerosols | 1% | Section 6.1 |
| albedo | 1% | Section 6 |
| Profile shape | 2% for elevation angles < 4°, negligible for higher elevation angles | Section 6.1 |
| RTM general | 1% | Wagner et al., 2019 |
| total | 4% | |
| | | |
| **$O_4$ VCD** | 2% | This study, section 5, see also Wagner et al., 2019 |

2. It would be interesting to have a time series of O4 dSCD/dAMF RMS values (similar to A13) for the data shown in Fig.6 and A11. I would expect a clear trend in the RMS differences over the day maybe similar to the one you showed for AOD and scaling factor? How is the correlation of these RMS values and the retrieved/measured AOD?

We prepared the requested figure:

[Figure]

The RMS values from the fit to the forward model show the same temporal trend as the RMS from MAPA, but the absolute values are smaller (as expected).

We also checked the correlation: No correlation was found (R²=0.00)

3. You mentioned that sun photometer measurements allow to differ between the aerosol particle size. Please show the contribution of differently sized aerosol particles to the total AOD over the day as well as all AODs and corresponding Angström exponents.

The following figure was added (new Fig. A3):

[Figure]

Fig. A3: Top: AOD at 500 nm attributed to the coarse and fine modes, as well as total AOD. Middle: Total AOD at different wavelengths. Bottom: Angström Exponents for selected wavelength pairs.

It should be noted that while creating these figures, it turned out that the AOD for the last measurement on 2 May 2019 (at 19:26) of this fully processed data set was about 30% higher than the value obtained from the initial AOD inversion (while all other measurements on that day were nearly identical). Even after consultation with the AERONET staff, no clear reason for this discrepancy could be identified. However, since the solar zenith angle is rather large (~84°), the extraction of the AOD, especially at short wavelengths is challenging, because of the strong Rayleigh

extinction. These uncertainties affect the comparison of the last elevation sequence (19:06 to 19:25).

We added the following information to the paper:

Section 2.2:

‚It should be noted that the uncertainties of the last AOD measurement on 2 May 2019, 19:26, are rather large because of the high SZA of 85°. In particular it was found that for that measurement the AOD from the fully processed sun photometer data (Fig. A3) was about 30% larger than the AOD of the initial retrieval (Fig. 3), while the results for all other measurements are almost identical. The radiative transfer simulations presented below for the last elevation sequence (19:06 to 19:25) are based on the initial (low) AOD values, which are in agreement with AOD measurements 20 minutes earlier. Nevertheless, the comparison results for this last elevation sequence should be treated with caution because of the large uncertainties of the corresponding AOD measurement.'

Figure caption of Fig. 7:

‚Note that for the last elevation sequence, the AOD used in the forward model has large uncertainties, see section 2.2.'

Figure caption of Fig. A12:

‚Note that for the last elevation sequence, the AOD used in the forward model has large uncertainties, see section 2.2.'

4. Furthermore, I was wondering why you only showed results for one day? The AODs for the following days are also rather small. Do these days support your findings?

The extremely low AODs only occurred on the selected day. Only at the beginning of the following day, still low AODs were measured (< 0.05 at 360 nm). However, during this period, the measurements at low elevation angles were strongly affected by clouds. Nevertheless, we compared the MAX-DOAS $O_4$ measurements retrieved during that period with radiative transfer simulations. Here, we only made simulations for an aerosol-free atmosphere to limit the effort (and also because of the rapid temporal variation of the AOD). The comparison results are shown below:

[Figure]

Fig. A13 Comparison of the measured and simulated $O_4$ dAMFs for two elevation sequences on 05 March 2019, when the AOD was rather small (<0.05 at 360 nm). The radiative transfer simulations were made for an aerosol-free atmosphere.

Like on 02 May 2019, the simulated $O_4$ dAMFs (for aerosol-free atmosphere) are smaller than the measurements (for cloud-free observations).

We added the following information to section 7.1.:
'It should be noted that that during the entire ship cruise, only during the beginning of 3 May 2019, similarly low (but still larger) AOD were measured as on 2 May 2019. We also compared the measured O4 dAMFs for the first two elevation sequences on 3 May to radiative transfer simulations. For that comparison we only made simulations for an aerosol-free atmosphere in order to limit the effort (and also because of the rapid temporal variation of the AOD during that time period). The results (see Fig. A13) are similar to those on 2 May 2019: except for the cloud contaminated measurements, the simulations are smaller than the measurements.'

Specific comments

P1, L22, 25: Please add a selection of corresponding references to the sentences starting with (L22) "In recent years,..." and (L25) "Several studies found that a scaling factor...".

We changed the text to:
'Several studies found that a scaling factor (SF<1) had to be applied to the observed atmospheric $O_4$ absorptions in order to bring them into agreement with radiative transfer simulations (e.g. Wagner et al., 2009; Clémer et al. 2010). Other studies, however, did not find the need to apply such a scaling factor (e.g. Spinei et al., 2015; Ortega et al., 2016). A more detailed discussion and overview on existing studies of both groups is provided in Wagner et al., 2019.'
We added the following references:

Spinei, E., Cede, A., Herman, J., Mount, G. H., Eloranta, E., Morley, B., Baidar, S., Dix, B., Ortega, I., Koenig, T., and Volkamer, R.: Ground-based direct-sun DOAS and airborne MAX-DOAS measurements of the collision-induced oxygen complex, O2O2, absorption with significant pressure and temperature differences, Atmos. Meas. Tech., 8, 793-809, https://doi.org/10.5194/amt-8-793-2015, 2015.

Wagner, T., Apituley, A., Beirle, S., Dörner, S., Friess, U., Remmers, J., and Shaiganfar, R.: Cloud detection and classification based on MAX-DOAS observations, Atmos. Meas. Tech., 7, 1289-1320, doi:10.5194/amt-7-1289-2014, 2014.

P2, Sec 2.2 and Fig.3: Since AODs at other wavelengths are available, please add them to Fig. 3.

We added a new figure (new Fig. A3) showing the AODs at all wavelengths, see reply to comment 3 above.

P3, L87: "(with..." => "(which...?
Corrected

Fig. A1: Your fit uses the wavelength range 352 - 387nm but Fig. A1 shows only wavelengths up to ~384nm. Please change the x-Axis according to the applied fitting window.

Many thanks for this hint!
In this study, we restricted the spectral range to 352 - 385 nm, because for some measurements (not on 2 May 2019) large spectral structures were found > 385 nm). However, for 2 May 2019, almost identical results (differences < 1%) were found for both spectral ranges.
We added this information to section 3.
The figure below shows the fit results for the spectral range 352 – 387 nm for the same spectrum as shown in Fig. A1. The results are almost identical to those shown in Fig. A1.

[Figure]

Fit results for a spectrum taken on 2 May 2019, 13:14:50, at an elevation angle of 1° (SZA: 33.6°). Left: results if a $H_2O$ cross section is included in the spectral analysis; right: results if no $H_2O$ cross section is included in the spectral analysis. The black lines represent the fitted cross section, the red lines indicate the residual (bottom) or the residual plus the fitted cross section.

Furthermore, I was wondering about the shown residual. It appears to me that there are still some residual structures left. Especially three peaks around 372-376 nm look familiar and could be attributed to Fraunhofer-Lines. Could this be somehow related to your Ring-treatment or do you have another explanation?

We have no plausible explanation for these structures. For the O4 analysis, these small remaining structures are not critical.

P5, L177: Why was the albedo set to 0.05? Please add a references here. As far as I know, we can expect a small dependence on SZA. How large is the impact on O4 when changing the albedo according to possible values?

We chose the value of 5% to be consistent with the MAPA inversions, and because it is appropriate for many parts of the global ocean. However, by having a closer look at maps of albedo (Kleipool et al., 2008) and chlorophyll content (e.g. from the NASA Earth Observatory: https://earthobservatory.nasa.gov/global-maps/MY1DMM_CHLORA), we found that at the specific location of the measurements, very clear waters exist, for which the surface albedo is typically higher (about 7 to 8%). The presence of very clear waters is also supported by the in situ chlorophyll measurements made aboard the ship.
We therefore made additional radiative transfer simulations using a surface albedo of 8%. We found that the obtained $O_4$ dAMFs were almost identical with those obtained for 5% surface albedo (differences <1%). The reason for the good agreement is that the effect of the surface albedo is similar for the $O_4$ AMFs for different elevation angles. Thus the effect of varying surface albedo almost cancels out.
We added this information to section 6.

P6, L215: "the the", please remove either first or second "the".

We removed both the first and second ‚the' and added a new one.

P6, L216: You wrote that Fig. A8 includes constant and linearly extrapolated values for lower altitudes but the greenish line does not look like a linear extrapolation to me. Why is that?

Many thanks for this hint. We corrected the figure.

P6, L221: Why is the Angström exponent "assumed" to be 2 when you have AODs at several wavelengths available to calculate more accurate values?

The AOD measurements represent the total AOD, but what is needed is the Angström exponent for the stratospheric aerosols. Therefore, the sun photometer measurements cannot be used for that purpose.

P6, L224: decsribed => described
Corrected

P7, L263: "smaller than" => "larger than"
Corrected

P7, L276: "bebetween" => "be between"
Corrected

Fig. A7 Could you please add a similar figure for the geometry with the smallest RAA to better assess the impact of AP and SSA variations throughout the day?

The figure for 13:30 (RAA ~0) is added.

---

## Author Comment (AC3) · 29 Mar 2021

**Reply to Referee #3**

First of all we want to thank this reviewer for the positive assessment and constructive comments.

We addressed these comments as explained in detail below.

The present manuscript addresses the issue of the difference between measured and simulated O4 dSCDS. Many studies over the last year used correction factors on the measured O4 dSCDs to achieve a better agreement without finding the physical explanation of these factors. Other studies support that the use of correction factor is not necessary. In previous studies, one possible explanation of this inconsistency was the uncertainties of aerosol information. It is very interesting that in this manuscript, this uncertainty is neglected because of the use of one day of measurements with very low AOD values.

I recommend the publication of the present manuscript. The content is clear, well explained and the manuscript falls into the scope of AMT.

**Many thanks for the positive assessment.**

Please consider some minor comments:

1. In Figure 1, I see that other days (or at least time windows during some days) have very small AOD values. Why these days are not included in your results? Would it be possible to include them and see if the results agree with the main findings of your study?

The extremely low AODs only occurred on the selected day. Only at the beginning of the following day, still low AODs were measured (

Fig. A13 Comparison of the measured and simulated  $O_4$  dAMFs for two elevation sequences on 05 March 2019, when the AOD was rather small (<0.05 at 360 nm). The radiative transfer simulations were made for an aerosol-free atmosphere.

Like on 02 May 2019, the simulated  $O_4$  dAMFs (for aerosol-free atmosphere) are smaller than the measurements (for cloud-free observations).

We added the following information to section 7.1.:

,It should be noted that that during the entire ship cruise, only during the beginning of 3 May 2019, similarly low (but still larger) AOD were measured as on 2 May 2019. We also compared the measured O4 dAMFs for the first two elevation sequences on 3 May to radiative transfer simulations. For that comparison we only made simulations for an aerosol-free atmosphere in order to limit the effort (and also because of the rapid temporal variation of the AOD during that time period). The results (see Fig. A13) are similar to those on 2 May 2019: except for the cloud contaminated measurements, the simulations are smaller than the measurements.'

2. The uncertainties that are described in the text are very important for your findings. That would be very useful if you could create a table of uncertainties.

A new table (table 3) was added to the paper (at the end of section 7.2):

[revised manuscript text omitted]

5. P. 4, Line 156 : Can you provide a possible explanation for this difference between ECMWF and in-situ measurements?

The most probable reasons for the discrepancies are originating from the rather coarse horizontal ( $\sim$ 80 km) and temporal (6 h) resolution of the ECMWF interim data set: First, the given model data is the average for the modelled box. Moreover, the simulation uncertainties are increased for parameterized subscale processes (e.g. wave motion) which do affect the in-situ measurements. This information is added to section 5.

6. P. 5, Line 175 : "simulations," instead of "simulations" Corrected

7. P. 5, Line 208, 210, Figure 4 : Is there any explanation why the raw data vary more with altitude? Is it valid to use the data above 3 km?

We added the following information to the figure caption: ,The scatter of the range corrected backscatter profiles increases, because the received raw signal scales with the inverse of the square of the distance.'

We used the profile data for altitudes, for which the signal stays positive (after smoothing). The exact altitude, at which the signal is set to zero has negligible influence on the simulated  $O_4$  dAMFs. We added this information to section 6.1.

8. Figure A8 : y axis varies from 0 km to 10 km and not from 0 km to 0 km. Please correct Corrected

9. P. 6, Line 221 : why the Angstrom coefficient is assumed equal to 2?

We took the value of 2 from existing publications (e.g. Malinina et al., 2019). However, most provided values are representative for larger wavelengths (typically 525 nm or larger). To estimate the uncertainties of the simulated  $O_4$  dSCDs related to

the uncertainty of the Angström exponent, we performed additional radiative transfer simulations assuming a stratospheric AOD of 0.008 (corresponding to an Angström exponent of 1). We found that the  $O_4$  dSCDs differ from those for a stratospheric AOD of 0.012 by less than 1%. We added this information to the paper.

10. P. 7, Line 276 : "be between" instead of "bebetween" Corrected

---

## Author Response (AR2)

Dear Andrew,

many thanks for accepting the paper, and for your suggestions about the frequent use of "it should be noted".
We followed your advice and removed this phrase at several locations in the text. At a few locations we found it still meaningful and kept it.

Best regards,

Thoams